# Efficient Multi-agent Reinforcement Learning by Planning

**Qihan Liu**[1*]**, Jianing Ye**[2*]**, Xiaoteng Ma**[1*]**, Jun Yang**[1†]**, Bin Liang**[1]**, Chongjie Zhang**[3†]

[1]Department of Automation, Tsinghua University
[2]Institute for Interdisciplinary Information Sciences, Tsinghua University
[3]Department of Computer Science & Engineering, Washington University in St. Louis
{lqh20, yejn21, ma-xt17}@mails.tsinghua.edu.cn
{yangjun603, bliang}@tsinghua.edu.cn
chongjie@wustl.edu

## Abstract

Multi-agent reinforcement learning (MARL) algorithms have accomplished remarkable breakthroughs in solving large-scale decision-making tasks. Nonetheless, most existing MARL algorithms are model-free, limiting sample efficiency and hindering their applicability in more challenging scenarios. In contrast, model-based reinforcement learning (MBRL), particularly algorithms integrating planning, such as MuZero, has demonstrated superhuman performance with limited data in many tasks. Hence, we aim to boost the sample efficiency of MARL by adopting model-based approaches. However, incorporating planning and search methods into multi-agent systems poses significant challenges. The expansive action space of multi-agent systems often necessitates leveraging the nearly-independent property of agents to accelerate learning. To tackle this issue, we propose the MAZero algorithm, which combines a centralized model with Monte Carlo Tree Search (MCTS) for policy search. We design a novel network structure to facilitate distributed execution and parameter sharing. To enhance search efficiency in deterministic environments with sizable action spaces, we introduce two novel techniques: Optimistic Search Lambda ($OS(\lambda)$) and Advantage-Weighted Policy Optimization (AWPO). Extensive experiments on the SMAC benchmark demonstrate that MAZero outperforms model-free approaches in terms of sample efficiency and provides comparable or better performance than existing model-based methods in terms of both sample and computational efficiency. Our code is available at https://github.com/liuqh16/MAZero.

## 1 Introduction

Multi-Agent Reinforcement Learning (MARL) has seen significant success in recent years, with applications in real-time strategy games (Arulkumaran et al., 2019; Ye et al., 2020), card games (Bard et al., 2020), sports games (Kurach et al., 2020), autonomous driving (Zhou et al., 2020), and multi-robot navigation (Long et al., 2018). Nonetheless, challenges within multi-agent environments have led to the problem of sample inefficiency. One key issue is the non-stationarity of multi-agent settings, where agents continuously update their policies based on observations and rewards, resulting in a changing environment for individual agents (Nguyen et al., 2020). Additionally, the joint action space's dimension can exponentially increase with the number of agents, leading to an immense policy search space (Hernandez-Leal et al., 2020). These challenges, combined with issues such as partial observation, coordination, and credit assignment, necessitate a considerable demand for samples in MARL for effective training (Gronauer & Diepold, 2022).

Conversely, Model-Based Reinforcement Learning (MBRL) has demonstrated its worth in terms of sample efficiency within single-agent RL scenarios, both in practice (Wang et al., 2019) and

---

[*]Equal contribution.
[†]Corresponding authors.

theory (Sun et al., 2019). Unlike model-free methods, MBRL approaches typically focus on learning a parameterized model that characterizes the transition or reward functions of the real environment (Sutton & Barto, 2018; Corneil et al., 2018; Ha & Schmidhuber, 2018). Based on the usage of the learned model, MBRL methods can be roughly divided into two lines: planning with model (Hewing et al., 2020; Nagabandi et al., 2018; Wang & Ba, 2019; Schrittwieser et al., 2020; Hansen et al., 2022) and data augmentation with model (Kurutach et al., 2018; Janner et al., 2019; Hafner et al., 2019; 2020; 2023). Due to the foresight inherent in planning methods and their theoretically guaranteed convergence properties, MBRL with planning often demonstrates significantly higher sample efficiency and converges more rapidly (Zhang et al., 2020). A well-known planning-based MBRL method is MuZero, which conducts Monte Carlo Tree Search (MCTS) with a value-equivalent learned model and achieves superhuman performance in video games like Atari and board games including Go, Chess and Shogi (Schrittwieser et al., 2020).

Despite the success of MBRL in single-agent settings, planning with the multi-agent environment model remains in its early stages of development. Recent efforts have emerged to bridge the gap by combining single-agent MBRL algorithms with MARL frameworks (Willemsen et al., 2021; Bargiacchi et al., 2021; Mahajan et al., 2021; Egorov & Shpilman, 2022). However, the extension of single-agent MBRL methods to multi-agent settings presents formidable challenges. On one hand, existing model designs for single-agent algorithms do not account for the unique biases inherent to multi-agent environments, such as the nearly-independent property of agents. Consequently, directly employing models from single-agent MBRL algorithms typically falls short in supporting efficient learning within real-world multi-agent environments, underscoring the paramount importance of model redesign. On the other hand, multi-agent environments possess action spaces significantly more intricate than their single-agent counterparts, thereby exponentially escalating the search complexity within multi-agent settings. This necessitates exploration into specialized planning algorithms tailored to these complex action spaces.

In this paper, we propose MAZero, the first empirically effective approach that extends the MuZero paradigm into multi-agent cooperative environments. In particular, our contributions are fourfold:

1. Inspired by the Centralized Training with Decentralized Execution (CTDE) concept, we develop a centralized-value, individual-dynamic model with shared parameters among all agents. We incorporate an additional communication block using the attention mechanism to promote cooperation during the model unrolling (see Section 4.1).

2. Given the deterministic characteristics of the learned model, we have devised the Optimistic Search Lambda (OS($\lambda$)) approach (see Section 4.2). It optimistically estimates the sampled returns while mitigating unrolling errors at larger depths using the parameter $\lambda$.

3. We propose a novel policy loss named Advantage-Weighted Policy Optimization (AWPO) (see Section 4.3) utilizing the value information calculated by OS($\lambda$) to improve the sampled actions.

4. We conduct extensive experiments on the SMAC benchmark, showing that MAZero achieves superior sample efficiency compared to model-free approaches and provides better or comparable performance than existing model-based methods in terms of both sample and computation efficiency (see Section 5).

## 2 BACKGROUND

This section introduces essential notations, and related works will be discussed in Appendix A.

**POMDP** Reinforcement Learning (RL) addresses the problem of an agent learning to act in an environment in order to maximize a scalar reward signal, which is characterized as a partially observable Markov decision process (POMDP) (Kaelbling et al., 1998) defined by $(\mathcal{S}, \mathcal{A}, T, U, \Omega, \mathcal{O}, \gamma)$, where $\mathcal{S}$ is a set of states, $\mathcal{A}$ is a set of possible actions, $T : \mathcal{S} \times \mathcal{A} \times \mathcal{S} \rightarrow [0, 1]$ is a transition function over next states given the actions at current states, $U : \mathcal{S} \times \mathcal{A} \rightarrow \mathbb{R}$ is the reward function. $\Omega$ is the set of observations for the agent and observing function $\mathcal{O} : \mathcal{S} \rightarrow \Omega$ maps states to observations. $\gamma \in [0, 1)$ is the discounted factor. At each timestep $t$, the agent acquire an observation $o_t = \mathcal{O}(s_t)$ based on current state $s_t$, choose action $a_t$ upon the history of observations $o_{\leq t}$ and receive corresponding reward $u_t = U(s_t, a_t)$ from the environment. The objective of the agent is to learn a policy $\pi$ that maximizes the expected discounted return $J(\pi) = \mathbb{E}_\pi \left[ \sum_{t=0}^{\infty} \gamma^t u_t | a_t \sim \pi(\cdot | o_{\leq t}) \right]$.

**MuZero**  MuZero (Schrittwieser et al., 2020) is an MBRL algorithm for single-agent settings that amalgamates a learned model of environmental dynamics with MCTS planning algorithm. MuZero's learned model consists of three functions: a *representation* function $h$, a *dynamics* function $g$, and a *prediction* function $f$. The model is conditioned on the observation history $o_{\leq t}$ and a sequence of future actions $a_{t:t+K}$ and is trained to predict rewards $r_{t,0:K}$, values $v_{t,0:K}$ and policies $p_{t,0:K}$, where $r_{t,k}, v_{t,k}, p_{t,k}$ are model predictions based on imaginary future state $s_{t,k}$ unrolling $k$ steps from current time $t$. Specifically, the *representation* function maps the current observation history $o_{\leq t}$ into a hidden state $s_{t,0}$, which is used as the root node of the MCTS tree. The *dynamic* function $g$ inputs the previous hidden state $s_{t,k}$ with an action $a_{t+k}$ and outputs the next hidden state $s_{t,k+1}$ and the predicted reward $r_{t,k}$. The *prediction* function $f$ computes the value $v_{t,k}$ and policy $p_{t,k}$ at each hidden state $s_{t,k}$. To perform MCTS, MuZero runs $N$ simulation steps where each consists of 3 stages: *Selection*, *Expansion* and *Backup*. During the *Selection* stage, MuZero starts traversing from the root node and selects action by employing the probabilistic Upper Confidence Tree (pUCT) rule (Kocsis & Szepesvári, 2006; Silver et al., 2016) until reaching a leaf node:

$$a = \arg\max_a \left[ Q(s,a) + P(s,a) \cdot \frac{\sqrt{\sum_b N(s,b)}}{1 + N(s,a)} \cdot c(s) \right] \tag{1}$$

where $Q(s,a)$ denotes the estimation for Q-value, $N(s,a)$ the visit count, $P(s,a)$ the prior probability of selecting action $a$ received from the *prediction* function, and $c(s)$ is used to control the influence of the prior relative to the Q-value. The target search policy $\pi(\cdot|s_{t,0})$ is the normalized distribution of visit counts at the root node $s_{t,0}$. The model is trained minimize the following 3 loss: reward loss $l_r$, value loss $l_v$ and policy loss $l_p$ between true values and network predictions.

$$L^{\text{MuZero}} = \sum_{k=0}^{K} [l_r(u_{t+k}, r_{t,k}) + l_v(z_{t+k}, v_{t,k}) + l_p(\pi_{t+k}, p_{t,k})] \tag{2}$$

where $u_{t+k}$ is the real reward from environment, $z_{t+k} = \sum_{i=0}^{n-1} \gamma^i u_{t+k+i} + \gamma^{t+k+n} \nu_{t+k+n}$ is n-step return consists of reward and MCTS search value, $\pi_{t+k}$ is the MCTS search policy.

EfficientZero (Ye et al., 2021) proposes an asynchronous parallel workflow to alleviate the computational demands of the reanalysis MCTS . EfficientZero incorporates the self-supervised consistency loss $l_s = \|s_{t,k} - s_{t+k,0}\|$ using the SimSiam network to ensure consistency between the hidden state from the $k$th step's dynamic $s_{t,k}$ and the direct representation of the future observation $s_{t+k,0}$.

**Sampled MuZero**  Since the efficiency of MCTS planning is intricately tied to the number of simulations performed, the application of MuZero has traditionally been limited to domains with relatively small action spaces, which can be fully enumerated on each node during the tree search. To tackle larger action spaces, Sampled MuZero (Hubert et al., 2021) introduces a sampling-based framework where policy improvement and evaluation are computed over small subsets of sampled actions. Specifically, every time in the *Expansion* stage, only a subset $T(s)$ of the full action space $\mathcal{A}$ is expanded, where $T(s)$ is on-time resampled from a policy $\beta$ base on the prior policy $\pi$. After that, in the *Selection* stage, an action is picked according to the modified pUCT formula.

$$a = \arg\max_{a \in T(s) \subset \mathcal{A}} \left[ Q(s,a) + \frac{\hat{\beta}}{\beta} P(s,a) \cdot \frac{\sqrt{\sum_b N(s,b)}}{1 + N(s,a)} \cdot c(s) \right] \tag{3}$$

where $\hat{\beta}$ is the empirical distribution of sampled actions, which means its support is $T(s)$.

In this way, when making actual decisions at some root node state $s_{t,0}$, Sampled MCTS will yield a policy $\omega(\cdot|s_{t,0})$ after $N$ simulations, which is a stochastic policy supported in $T(s_{t,0})$. The actual improved policy is denoted as $\pi^{\text{MCTS}}(\cdot|s_{t,0}) = \mathbb{E}[\omega(\cdot|s_{t,0})]$, which is the expectation of $\omega(\cdot|s_{t,0})$. The randomness that the expectation is taken over is from all sampling operations in Sampled MCTS. As demonstrated in the original paper, Sampled MuZero can be seamlessly extended to multi-agent settings by considering each agent's policy as a distinct dimension for a single agent's multi-discrete action space during centralized planning. The improved policy is denoted as $\boldsymbol{\pi}^{\text{MCTS}}(\cdot|\boldsymbol{s}_{t,0})$.[1]

---

[1]For the sake of simplicity, we will not emphasize the differences between search policy $\boldsymbol{\pi}^{\text{MCTS}}(\cdot|\boldsymbol{s}_{t,0})$ in multi-agent settings and $\pi^{\text{MCTS}}(\cdot|s_{t,0})$ in single-agent settings apart from the mathematical expressions.

## 3 Challenges in Planning-based Multi-agent Model-based RL

Extending single-agent Planning-based MBRL methods to multi-agent environments is highly non-trivial. On one hand, existing model designs for single-agent algorithms do not account for the unique biases inherent to multi-agent environments, such as the nearly-independent property of agents. This renders the direct employing a flattened model from single-agent MBRL algorithms typically falls short in supporting efficient learning within real-world multi-agent environments. On the other hand, multi-agent environments possess state-action spaces significantly more intricate than their single-agent counterparts, thereby exponentially escalating the search complexity within multi-agent settings. This, in turn, compels us to explore specialized search algorithms tailored to complex action spaces. Moreover, the form of the model, in tandem with its generalization ability, collectively constrains the form and efficiency of the search algorithm, and vice versa, making the design of the model and the design of the search algorithm highly interrelated.

In this section, we shall discuss the challenges encountered in the current design of planning-based MBRL algorithms, encompassing both model design and searching algorithm aspects. We will elucidate how our MAZero algorithm systematically addresses these two issues in the Section 4.

### 3.1 Model Design

Within the paradigm of centralized training with decentralized execution (CTDE), one straightforward approach is to learn a joint model that enables agents to do centralized planning within the joint policy space. However, given the large size of the state-action space in multi-agent environments, the direct application of a flattened model from single-agent MBRL algorithms tends to render the learning process inefficient (Figure 6). This inefficiency stems from the inadequacy of a flattened model in accommodating the unique biases inherent in multi-agent environments.

In numerous real-world scenarios featuring multi-agent environments, agents typically engage in independent decision-making for the majority of instances, with collaboration reserved for exceptional circumstances. Furthermore, akin agents often exhibit homogeneous behavior (e.g., focusing fire in the SMAC environment). For the former, a series of algorithms, including IPPO (Yu et al., 2022), have demonstrated the efficacy of independent learning in a multitude of MARL settings. Regarding the latter, previous research in multi-agent model-free approaches has underscored the huge success of parameter-sharing (Rashid et al., 2020; Sunehag et al., 2017) within MARL, which can be regarded as an exploitation of agents' homogenous biases. Hence, encoding these biases effectively within the design of the model becomes one main challenge in multi-agent MBRL.

### 3.2 Planning in Exponential Sized Action Space

The action space in multi-agent environments grows exponentially with the number of agents, rendering it immensely challenging for vanilla MCTS algorithms to expand all actions. While Sampled MuZero, in prior single-agent MBRL work, attempted to address scenarios with large action spaces by utilizing action sampling, the size of the environments it tackled (e.g., *Go*, $\sim$ 300 actions) still exhibits substantial disparity compared to typical multi-agent environments (e.g., *SMAC*-`27m_vs_30m`, $\sim 10^{42}$ actions).

In practical applications like MARL, due to the significantly fewer samples taken compared to the original action space size, the underestimation issue of Sampled MCTS becomes more pronounced, making it less than ideal in terms of searching efficiency. This motivates us to design a more optimistic search process.

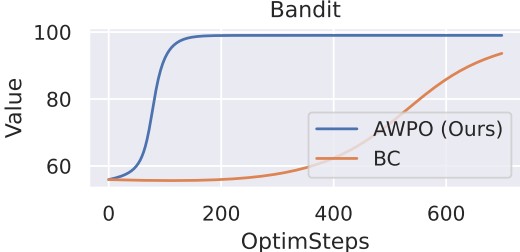

Figure 1: **Bandit Experiment** We compare the Behavior Cloning (BC) loss and our Advantage-Weighted Policy Optimization (AWPO) loss on a bandit with action space $|\mathcal{A}| = 100$ and sampling time $B = 2$. It is evident that AWPO converges much faster than BC, owing to the effective utilization of values.

Moreover, the behavior cloning (BC) loss that Sampled MCTS algorithms adopt disregards the value information. This is because the target policy $\omega(\cdot|s_{t,0})$ only encapsulates the relative magnitude of sampled action values while disregarding the information of the absolute action values. The issue is not particularly severe when dealing with a smaller action space. However, in multi-agent environments, the disparity between the sampling time $B$ and $|\mathcal{A}|$ becomes significant, making it impossible to overlook the repercussions of disregarding value information (Figure 1).

## 4 MAZERO ALGORITHM

### 4.1 MODEL STRUCTURE

The MAZero model comprises 6 key functions: a *representation* function $h_\theta$ for mapping the current observation history $o^i_{\leq t}$ of agent $i$ to an individual latent state $s^i_{t,0}$, a *communication* function $e_\theta$ which generates additional cooperative features $e^i_{t,k}$ for each agent $i$ through the attention mechanism, a *dynamic* function $g_\theta$ tasked with deriving the subsequent local latent state $s^i_{t,k+1}$ based on the agent's individual state $s^i_{t,k}$, future action $a^i_{t+k}$ and communication feature $e^i_{t,k}$, a *reward prediction* function forecasting the cooperative team reward $r_{t,k}$ from the global hidden state $\boldsymbol{s}_{t,k} = (s^1_{t,k}, \ldots, s^N_{t,k})$ and joint action $\boldsymbol{a}_{t+k} = (a^1_{t+k}, \ldots, a^N_{t+k})$, a *value prediction* function $V_\theta$ aimed at predicting value $v_{t,k}$ for each global hidden state $\boldsymbol{s}_{t,k}$, and a *policy prediction* function $P_\theta$ which given an individual state $s^i_{t,k}$ and generates the corresponding policy $p^i_{t,k}$ for agent $i$. The model equations are shown in Equation (4).

$$\begin{cases} \text{Representation:} & s^i_{t,0} = h_\theta(o^i_{\leq t}) \\ \text{Communication:} & e^1_{t,k}, \ldots, e^N_{t,k} = e_\theta(s^1_{t,k}, \ldots, s^N_{t,k}, a^1_{t+k}, \ldots, a^N_{t+k}) \\ \text{Dynamic:} & s^i_{t,k+1} = g_\theta(s^i_{t,k}, a^i_{t+k}, e^i_{t,k}) \\ \text{Reward Prediction:} & r_{t,k} = R_\theta(s^1_{t,k}, \ldots, s^N_{t,k}, a^1_{t+k}, \ldots, a^N_{t+k}) \\ \text{Value Prediction:} & v_{t,k} = V_\theta(s^1_{t,k}, \ldots, s^N_{t,k}) \\ \text{Policy Prediction:} & p^i_{t,k} = P_\theta(s^i_{t,k}) \end{cases} \tag{4}$$

Notably, the representation function $h_\theta$, the dynamic function $g_\theta$, and the policy prediction function $P_\theta$ all operate using local information, enabling distributed execution. Conversely, the other functions deal with value information and team cooperation, necessitating centralized training for effective learning.

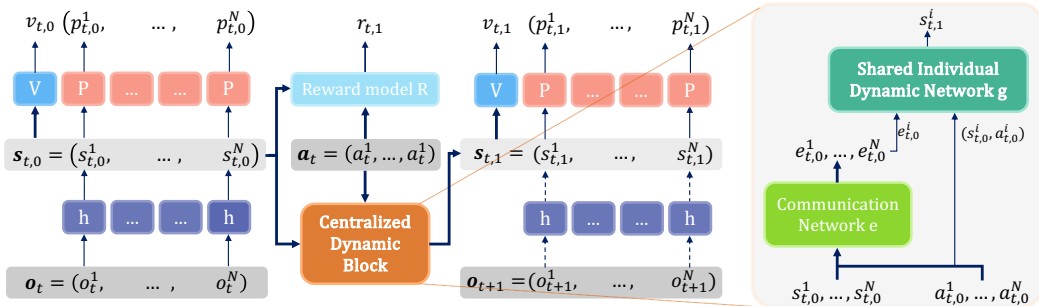

Figure 2: **MAZero model structure** Given the current observations $o^i_t$ for each agent, the model separately maps them into local hidden states $s^i_{t,0}$ using a shared representation network $h$. Value prediction $v_{t,0}$ is computed based on the global hidden state $\boldsymbol{s}_{t,0}$ while policy priors $p^i_{t,0}$ are individually calculated for each agent using their corresponding local hidden states. Agents use the communication network $e$ to access team information $e^i_{t,0}$ and generate next local hidden states $s^i_{t,1}$ via the shared individual dynamic network $g$, subsequently deriving reward $r_{t,1}$, value $v_{t,1}$ and policies $p^i_{t,1}$. During the training stage, real future observations $\boldsymbol{o}_{t+1}$ can be obtained to generate the target for the next hidden state, denoted as $\boldsymbol{s}_{t+1,0}$.

### 4.2 Optimistic Search Lambda

Having discerned that MAZero has acquired a deterministic world model, we devise a Optimistic Search Lambda (OS($\lambda$)) approach to better harness this characteristic of the model.

In previous endeavors, the selection stage of MCTS employed two metrics (value score and exploration bonus) to gauge our interest in a particular action. In this context, the value score utilized the mean estimate of values obtained from all simulations within that node's subtree. However, within the deterministic model, the mean estimate appears excessively conservative. Contemplating a scenario where the tree degenerates into a multi-armed bandit, if pulling an arm consistently yields a deterministic outcome, there arises no necessity, akin to the Upper Confidence Bound (UCB) algorithm, to repeatedly sample the same arm and calculate its average. Similarly, within the tree-based version of the UCT (Upper Confidence Trees) algorithm, averaging values across all simulations in a subtree may be substituted with a more optimistic estimation when the environment is deterministic.

Hence, courtesy of the deterministic model, our focus narrows to managing model generalization errors rather than contending with errors introduced by environmental stochasticity. Built upon this conceptual foundation, we have devised a methodology for calculating the value score that places a heightened emphasis on optimistic values.

Specifically, for each node $s$, we define:

$$\mathcal{U}_d(s) = \left\{ \sum_{k<d} \gamma^k r_k + \gamma^d v(s') \, \middle| \, s' : \text{dep}(s') = \text{dep}(s) + d \right\} \tag{5}$$

$$\mathcal{U}_d^\rho(s) = \text{Top } (1 - \rho) \text{ values in } \mathcal{U}_d \tag{6}$$

$$V_\lambda^\rho(s) = \sum_d \sum_{x \in \mathcal{U}_d^\rho(s)} \lambda^d x \Big/ \sum_d \lambda^d |\mathcal{U}_d^\rho(s)| \tag{7}$$

$$A_\lambda^\rho(s, a) = r(s, a) + \gamma V_\lambda^\rho(\text{Dynamic}(s, a)) - v(s) \tag{8}$$

where $\text{dep}(u)$ denotes the depth of node $u$ in the MCTS tree, $\rho, \lambda \in [0, 1]$ are hyperparameters, $r, v$, and Dynamic are all model predictions calculated according to Equation (4).

To offer a brief elucidation on this matter, Equation (6) is the set of optimistic values at depth $d$, the degree of optimism is controlled by the quantile parameter $\rho$. Since the model errors magnify with increasing depth in practice, Equation (7) calculates a weighted mean of all optimistic values in the subtree as the value estimation of $s$, where the weight of different estimations are discounted by a factor $\lambda$ over the depth.

Finally, we use the optimistic advantage (Equation (8)) to replace the value score in Equation (3) for optimistic search, that is:

$$a = \underset{a \in T(s) \subset \mathcal{A}}{\arg\max} \left[ A_\lambda^\rho(s, a) + \frac{\hat{\beta}}{\beta} P(s, a) \cdot \frac{\sqrt{\sum_b N(s, b)}}{1 + N(s, a)} \cdot c(s) \right] \tag{9}$$

### 4.3 Advantage-Weighted Policy Optimization

In order to utilize the value information calculated by OS($\lambda$), we propose Advantage-Weighted Policy Optimization (AWPO), a novel policy loss function that incorporates the optimistic advantage (Equation (8)) into the behavior cloning loss.[2]

In terms of outcome, AWPO loss weights the per-action behavior cloning loss by $\exp\left(\frac{A_\lambda^\rho(s,a)}{\alpha}\right)$:

$$l_p(\theta; \pi, \omega, A_\lambda^\rho) = -\mathbb{E}\left[ \sum_{a \in T(s)} \omega(a|s) \exp\left( \frac{A_\lambda^\rho(s, a)}{\alpha} \right) \log \pi(a|s; \theta) \right] \tag{10}$$

---

[2]In this section, the expectation operator is taken over the randomness of OS($\lambda$). For the sake of simplicity, we will omit this in the formula.

where $\theta$ denotes parameters of the learned model, $\pi(\cdot|s;\theta)$ is the network predict policy to be improved, $\omega(\cdot|s)$ is the search policy supported in action subset $T(s)$, $A_\lambda^\rho$ is the optimistic advantage derived from OS($\lambda$) and $\alpha > 0$ is a hyperparameter controlling the degree of optimism. Theoretically, AWPO can be regarded as a cross-entropy loss between $\eta^*$ and $\pi$,[3] where $\eta^*$ is the non-parametric solution of the following constrained optimization problem:

$$
\begin{aligned}
\underset{\eta}{\text{maximize}} \quad & \mathbb{E}_{a \sim \eta(\cdot|s)}\left[A_\lambda^\rho(s,a)\right] \\
\text{s.t.} \quad & \text{KL}\left(\eta(\cdot|s)\|\pi^{\text{MCTS}}(\cdot|s)\right) \leq \epsilon
\end{aligned}
\tag{11}
$$

To be more specific, by Lagrangian method, we have:

$$
\eta^*(a|s) \propto \pi^{\text{MCTS}}(a|s) \exp\left(\frac{A_\lambda^\rho(s,a)}{\alpha}\right) = \mathbb{E}\left[\omega(a|s)\exp\left(\frac{A_\lambda^\rho(s,a)}{\alpha}\right)\right]
\tag{12}
$$

Thereby, minimizing the KL divergence between $\eta^*$ and $\pi$ gives the form of AWPO loss:

$$
\begin{aligned}
& \arg\min_\theta \text{KL}(\eta^*(\cdot|s)\|\pi(\cdot|s;\theta)) \\
=\; & \arg\min_\theta -\frac{1}{Z(s)}\sum_a \pi^{\text{MCTS}}(a|s)\exp\left(\frac{A_\lambda^\rho(s,a)}{\alpha}\right)\log\pi(a|s;\theta) - H(\eta^*(\cdot|s)) \\
=\; & \arg\min_\theta l_p(\theta;\pi,\omega,A_\lambda^\rho)
\end{aligned}
$$

where $Z(s) = \sum_a \pi^{\text{MCTS}}(a|s)\exp\left(\frac{A_\lambda^\rho(s,a)}{\alpha}\right)$ is a normalizing factor, $H(\eta^*)$ stands for the entropy of $\eta^*$, which is a constant.

Equation (11) shows that AWPO aims to optimize the value improvement of $\pi$ in proximity to $\pi^{\text{MCTS}}$, which effectively combines the improved policy obtained from OS($\lambda$) with the optimistic value, thereby compensating for the shortcomings of BC loss and enhancing the efficiency of policy optimization.

Details of the derivation can be found in Appendix G.

## 4.4 Model Training

The MAZero model is unrolled and trained in an end-to-end schema similar to MuZero. Specifically, given a trajectory sequence of length $K+1$ for observation $\boldsymbol{o}_{t:t+K}$, joint actions $\boldsymbol{a}_{t:t+K}$, rewards $u_{t:t+K}$, value targets $z_{t:t+K}$, policy targets $\boldsymbol{\pi}_{t:t+K}^{\text{MCTS}}$ and optimistic advantages $A_\lambda^\rho$, the model is unrolled for $K$ steps as is shown in Figure 2 and is trained to minimize the following loss:

$$
\mathcal{L} = \sum_{k=1}^{K}(l_r + l_v + l_s) + \sum_{k=0}^{K} l_p
\tag{13}
$$

where $l_r = \|r_{t,k} - u_{t+k}\|$ and $l_v = \|v_{t,k} - z_{t+k}\|$ are reward and value losses similar to MuZero, $l_s = \|\boldsymbol{s}_{t,k} - \boldsymbol{s}_{t+k,0}\|$ is the consistency loss akin to EfficientZero and $l_p$ is the AWPO policy loss (Equation (10)) under the realm of multi-agent joint action space:

$$
l_p = -\sum_{\boldsymbol{a}\in T(\boldsymbol{s}_{t+k,0})} \omega(\boldsymbol{a}|\boldsymbol{s}_{t+k,0})\exp\left(\frac{A_\lambda^\rho(\boldsymbol{s}_{t+k,0},\boldsymbol{a})}{\alpha}\right)\log\boldsymbol{\pi}(\boldsymbol{a}|\boldsymbol{s}_{t,k};\theta)
\tag{14}
$$

where OS($\lambda$) is performed under the hidden state $\boldsymbol{s}_{t+k,0}$ directly represented from the actual future observation $\boldsymbol{o}_{t+k}$, deriving corresponding action subset $T(\boldsymbol{s}_{t+k,0})$, search policy $\omega(\boldsymbol{a}|\boldsymbol{s}_{t+k,0})$ and optimistic advantage $A_\lambda^\rho(\boldsymbol{s}_{t+k,0},\boldsymbol{a})$. $\boldsymbol{\pi}(\boldsymbol{a}|\boldsymbol{s}_{t,k};\theta) = \prod_{i=1}^{N}P_\theta(a^i|s_{t,k}^i)$ denotes the joint policy to be improved at the $k$th step's hidden state $\boldsymbol{s}_{t,k}$ unrolling from dynamic function.

---

[3]It is equivalent to minimizing the KL divergence of $\eta^*$ and $\pi$.

## 5 EXPERIMENTS

### 5.1 STARCRAFT MULTI-AGENT CHALLENGE

**Baselines** We compare MAZero with both model-based and model-free baseline methods on StarCraft Multi-Agent Challenge (SMAC) environments. The model-based baseline is MAMBA (Egorov & Shpilman, 2022), a recently introduced multi-agent MARL algorithm based on DreamerV2 (Hafner et al., 2020) known for its state-of-the-art sample efficiency in various SMAC tasks. The model-free MARL methods includes QMIX (Rashid et al., 2020), QPLEX (Wang et al., 2020a), RODE (Wang et al., 2020b), CDS (Li et al., 2021), and MAPPO (Yu et al., 2022).

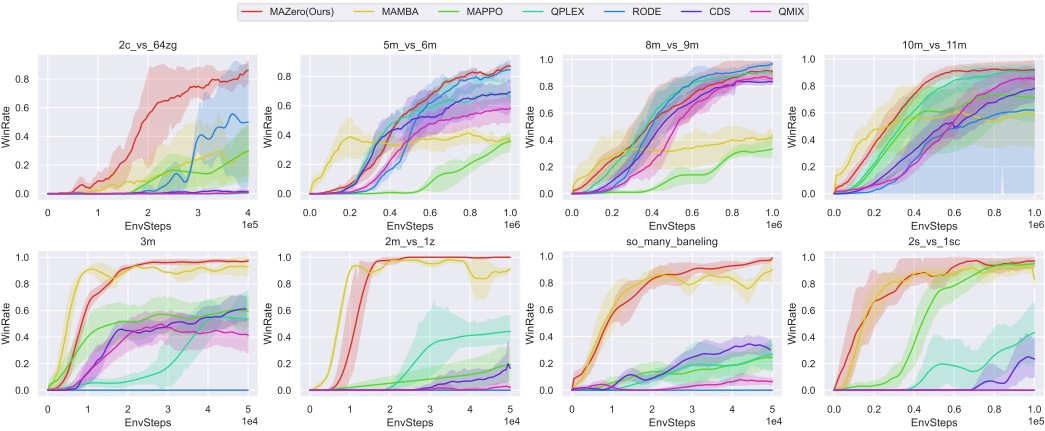

Figure 3: Comparisons against baselines in SMAC. Y axis denotes the win rate and X axis denotes the number of steps taken in the environment. Each algorithm is executed with 10 random seeds.

Figure 3 illustrates the results in the SMAC environments. Among the 23 available scenarios, we have chosen 8 for presentation in this paper. Specifically, we have selected four random scenarios categorized as *Easy* tasks and 4 *Hard* tasks. It is evident that MAZero outperforms all baseline algorithms across eight scenarios with a given number of steps taken in the environment. Notably, both MAZero and MAMBA, which are categorized as model-based methods, exhibit markedly superior sample efficiency in easy tasks when compared to model-free baselines. Furthermore, MAZero displays a more stable training curve and smoother win rate during evaluation than MAMBA. In the realm of hard tasks, MAZero surpasses MAMBA in terms of overall performance, with a noteworthy emphasis on the *2c_vs_64zg* scenario. In this particular scenario, MAZero attains a higher win rate with a significantly reduced sample size when contrasted with other methods. This scenario involves a unique property, featuring only two agents and an expansive action space of up to 70 for each agent, in contrast to other scenarios where the agent's action space is generally fewer than 20. This distinctive characteristic enhances the role of planning in MAZero components, similar to MuZero's outstanding performance in the domain of Go.

As MAZero builds upon the MuZero framework, we perform end-to-end training directly using planning results from the replay buffer. Consequently, this approach circumvents the time overhead for data augmentation, as seen in Dreamer-based methods. Figure 4 illustrates the superior performance of MAZero with respect to the temporal cost in SMAC environments when compared to MAMBA.

### 5.2 ABLATION

We perform several ablation experiments on the two proposed techniques: OS($\lambda$) and AWPO. The results with algorithms executed with three random seeds are reported in Figure 5. In particular, we examine whether disabling OS($\lambda$), AWPO, or both of them (i.e., using original Sampled MCTS and BC loss function) impairs final performance. Significantly, both techniques greatly impact final performance and learning efficiency.

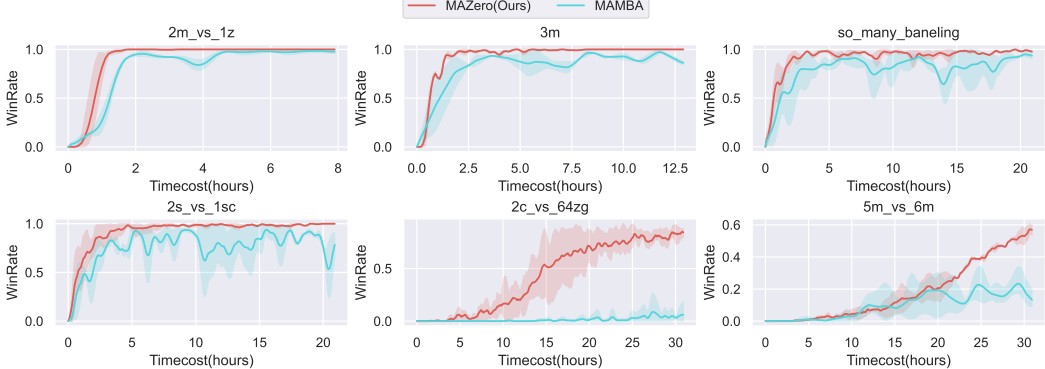

Figure 4: Comparisons against MBRL baselines in SMAC. The y-axis denotes the win rate, and the X-axis denotes the cumulative run time of algorithms in the same platform.

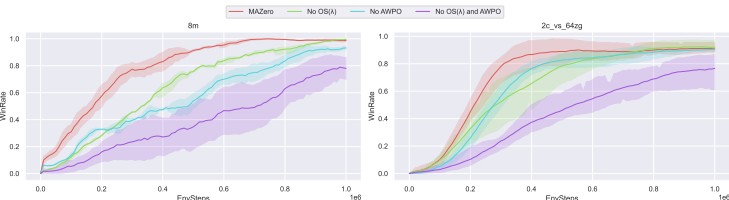

Figure 5: Ablation study on proposed approaches for planning.

In our ablation of the MAZero network structure (Figure 6), we have discerned the substantial impact of two components, "communication" and "sharing", on algorithmic performance. In stark contrast, the flattened model employed in single-agent MBRL, due to its failure to encapsulate the unique biases of multi-agent environments, can only learn victorious strategies in the most rudimentary of scenarios, such as the *2m_vs_1z* map.

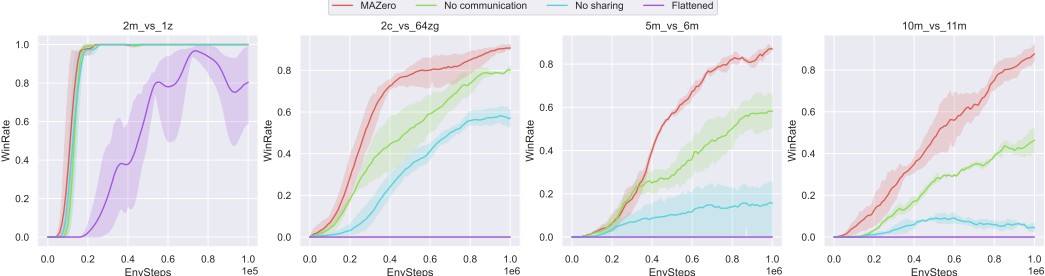

Figure 6: Ablation study on network structure.

# 6 CONCLUSION

In this paper, we introduce a model-based multi-agent algorithm, MAZero, which utilizes the CTDE framework and MCTS planning. This approach boasts superior sample efficiency compared to state-of-the-art model-free methods and provides comparable or better performance than existing model-based methods in terms of both sample and computational efficiency. We also develop two novel approaches, OS($\lambda$) and AWPO, to improve search efficiency in vast action spaces based on sampled MCTS. In the future, we aim to address this issue through reducing the dimensionality of the action space in search, such as action representation.

## 7 REPRODUCIBILITY

We ensure the reproducibility of our research by providing comprehensive information and resources that allow others to replicate our findings. All experimental settings in this paper are available in Appendix B, including details on environmental setup, network structure, training procedures, hyperparameters, and more. The source code utilized in our experiments along with clear instructions on how to reproduce our results will be made available upon finalization of the camera-ready version. The benchmark employed in this investigation is either publicly accessible or can be obtained by contacting the appropriate data providers. Additionally, detailed derivations for our theoretical claims can be found in Appendix G. We are dedicated to addressing any concerns or inquiries related to reproducibility and are open to collaborating with others to further validate and verify our findings.

ACKNOWLEDGMENTS

This work was supported by the National Science and Technology Innovation 2030 - Major Project (Grant No. 2022ZD0208804).

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

## A    RELATED WORKS

**Dec-POMDP**    In this work, we focus on the fully cooperative multi-agent systems that can be formalized as Decentralized Partially Observable Markov Decision Process (Dec-POMDP) (Oliehoek et al., 2016), which are defined by $(N, \mathcal{S}, \mathcal{A}^{1:N}, T, U, \Omega^{1:N}, \mathcal{O}^{1:N}, \gamma)$, where $N$ is the number of agents, $\mathcal{S}$ is the global state space, $T$ a global transition function, $U$ a shared reward function and $\mathcal{A}^i, \Omega^i, \mathcal{O}^i$ are the action space, observation space and observing function for agent $i$. Given state $s_t$ at timestep $t$, agent $i$ can only acquire local observation $o_t^i = \mathcal{O}^i(s_t)$ and choose action $a_t^i \in A^i$ according to its policy $\pi^i$ based on local observation history $o_{\leq t}^i$. The environment then shifts to the next state $s_{t+1} \sim T(\cdot|s_t, \boldsymbol{a}_t)$ and returns a scalar reward $u_t = U(s_t, \boldsymbol{a}_t)$. The objective for all agents is to learn a joint policy $\boldsymbol{\pi}$ that maximizes the expectation of discounted return $J(\boldsymbol{\pi}) = \mathbb{E}_{\boldsymbol{\pi}} \left[ \sum_{t=0}^{\infty} \gamma^t u_t | a_t^i \sim \pi^i(\cdot|o_{\leq t}^i), i = 1, \ldots, N \right]$.

**Combining reinforcement learning and planning algorithms**    The integration of reinforcement learning and planning within a common paradigm has yielded superhuman performance in diverse domains (Tesauro, 1994; Silver et al., 2017; Anthony et al., 2017; Silver et al., 2018; Schrittwieser et al., 2020; Brown et al., 2020). In this approach, RL acquires knowledge through the learning of value and policy networks, which in turn guide the planning process. Simultaneously, planning generates expert targets that facilitate RL training. For example, MuZero-based algorithms (Schrittwieser et al., 2020; Hubert et al., 2021; Antonoglou et al., 2021; Ye et al., 2021; Mei et al., 2022) combine Monte-Carlo Tree Search (MCTS), TD-MPC (Hansen et al., 2022) integrates Model Predictive Control (MPC), and ReBeL (Brown et al., 2020) incorporate a fusion of Counterfactual Regret Minimization (CFR). Nevertheless, the prevailing approach among these algorithms typically involves treating the planning results as targets and subsequently employing behavior cloning to enable the RL network to emulate these targets. Muesli (Hessel et al., 2021) first combines regularized policy optimization with model learning as an auxiliary loss within the MuZero framework. However, it focuses solely on the policy gradient loss while neglecting the potential advantages of planning. Consequently, it attains at most a performance level equivalent to that of MuZero. To the best of our knowledge, we are the first to combined policy gradient and MCTS planning to accurate policy learning in model-based reinforcement learning.

**Variants of MuZero-based algorithms**    While the original MuZero algorithm (Schrittwieser et al., 2020) has achieved superhuman performance levels in Go and Atari, it is important to note that it harbors several limitations. To address and overcome these limitations, subsequent advancements have been made in the field. EfficientZero (Ye et al., 2021), for instance, introduces self-supervised consistency loss, value prefix prediction, and off-policy correction mechanisms to expedite and stabilize the training process. Sampled MuZero (Hubert et al., 2021) extends its capabilities to complex action spaces through the incorporation of sample-based policy iteration. Gumbel MuZero (Danihelka et al., 2021) leverages the Gumbel-Top-k trick and modifies the planning process using Sequential Halving, thereby reducing the demands of MCTS simulations. SpeedyZero (Mei et al., 2022) integrates a distributed RL framework to facilitate parallel training, further enhancing training efficiency. He et al. (2023) demonstrate that the model acquired by MuZero typically lacks accuracy when employed for policy evaluation, rendering it ineffective in generalizing to assess previously unseen policies.

**Multi-agent reinforcement learning**    The typical approach for MARL in cooperative settings involves centralized training and decentralized execution (CTDE). During the training phase, this approach leverages global or communication information, while during the execution phase, it restricts itself to the observation information relevant to the current agent. This paradigm encompasses both value-based (Sunehag et al., 2017; Rashid et al., 2020; Son et al., 2019; Yang et al., 2020; Wang et al., 2020a) and policy-based (Lowe et al., 2017; Liu et al., 2020; Peng et al., 2021; Ryu et al., 2020; Ye et al., 2023) MARL methods within the context of model-free scenarios. Model-based MARL algorithms remains relatively underexplored with only a few methods as follows: MAMBPO (Willemsen et al., 2021) pioneers the fusion of the CTDE framework with Dyan-style model-based techniques, emphasizing the utilization of generated data closely resembling real-world data. CPS (Bargiacchi et al., 2021) introduces dynamic and reward models to determine data priorities based on the MAMBPO approach. Tesseract (Mahajan et al., 2021) dissects states and actions into low-rank tensors and evaluates the Q function using Dynamic Programming (DP) within an approximate en-

vironment model. MAMBA (Egorov & Shpilman, 2022) incorporates the word model proposed in DreamerV2 (Hafner et al., 2020) and introduces an attention mechanism to facilitate communication. This integration leads to noteworthy performance improvements and a substantial enhancement in sample efficiency when compared to previous model-free methods. To the best of our knowledge, we are the first to expand the MuZero framework and incorporate MCTS planning into the context of model-based MARL settings.

## B  IMPLEMENTATION DETAILS

### B.1  NETWORK STRUCTURE

For the SMAC scenarios where the input observations are 1 dimensional vectors (as opposed to 2 dimensional images for board games or Atari used by MuZero), we use a variation of the MuZero model architecture in which all convolutions are replaced by fully connected layers. The model consists of 6 modules: representation function, communication function, dynamic function, reward prediction function, value prediction function, and policy prediction function, which are all represented as neural networks. All the modules except the communication block are implemented as Multi-Layer Perception (MLP) networks, where each liner layer in MLP is followed by a Rectified Linear Unit (ReLU) activation and a Layer Normalisation (LN) layer. Specifically, we use a hidden state size of 128 for all SMAC scenarios, and the hidden layers for each MLP module are as follows: Representation Network = $[128, 128]$, Dynamic Network = $[128, 128]$, Reward Network = $[32]$, Value Network = $[32]$ and Policy Network = $[32]$. We use Transformer architecture (Vaswani et al., 2017) with three stacked layers for encoding state-action pairs. These encodings are then used by the agents to process local dynamic and make predictions. We use a dropout technique with probability $p = 0.1$ to prevent the model from over-fitting and use positional encoding to distinguish agents in homogeneous settings.

For the representation network, we stack the last four local observations together as input for each agent to deal with partial observability. Additionally, the concatenated observations are processed by an extra LN to normalize observed features before representation.

The dynamic function first concatenates the current local state, the individual action, and the communication encoding based on current state-action pairs as input features. To alleviate the problem that gradients tend to zero during the continuous unroll of the model, the dynamic function employs a residual connection between the next hidden state and the current hidden state.

For value and reward prediction, we follow in scaling targets using an invertible transform $h(x) = sign(x)\sqrt{1+x} - 1 + 0.001 \cdot x$ and use the categorical representation introduced in MuZero (Schrittwieser et al., 2020). We use ten bins for both the value and the reward predictions, with the predictions being able to represent values between $[-5, 5]$. We used $n-$step bootstrapping with $n = 5$ and a discount of $0.99$.

### B.2  TRAINING DETAILS

MAZero employs a similar pipeline to EfficientZero (Ye et al., 2021) while asynchronizing the parallel stages of data collection (Self-play workers), reanalysis (Reanalyze workers), and training (Main Thread) to maintain the reproducibility of the same random seed.

We use the Adam optimizer (Kingma & Ba, 2014) for training, with a batch size of 256 and a constant learning rate of $10^{-4}$. Samples are drawn from the replay buffer according to prioritized replay (Schaul et al., 2015) using the same priority and hyperparameters as in MuZero.

In practice, we re-execute OS($\lambda$) using a delayed target model $\hat{\theta}$ in the *reanalysis* stage to reduce off-policy error. Consequently, the AWPO loss function actually used is the following equation:

$$l_p = - \sum_{\hat{\boldsymbol{a}} \in \hat{T}(\hat{\boldsymbol{s}}_{t+k,0})} \hat{\omega}(\hat{\boldsymbol{a}}|\hat{\boldsymbol{s}}_{t+k,0}) \exp\left(\frac{\hat{A}_\lambda^\rho(\hat{\boldsymbol{s}}_{t+k,0}, \hat{\boldsymbol{a}})}{\alpha}\right) \log \boldsymbol{\pi}(\hat{\boldsymbol{a}}|\boldsymbol{s}_{t,k}; \theta)$$

For other details, we provide hyper-parameters in Table 1.

| Parameter | Setting |
|---|---|
| Observations stacked | 4 |
| Discount factor | 0.99 |
| Minibatch size | 256 |
| Optimizer | Adam |
| Optimizer: learning rate | $10^{-4}$ |
| Optimizer: RMSprop epsilon | $10^{-5}$ |
| Optimizer: weight decay | 0 |
| Max gradient norm | 5 |
| Priority exponent($c_\alpha$) | 0.6 |
| Priority correction($c_\beta$) | $0.4 \rightarrow 1$ |
| Evaluation episodes | 32 |
| Min replay size for sampling | 300 |
| Target network updating interval | 200 |
| Unroll steps | 5 |
| TD steps($n$) | 5 |
| Number of MCTS sampled actions($K$) | 10 |
| Number of MCTS simulations($N$) | 100 |
| Quantile in MCTS value estimation($\rho$) | 0.75 |
| Decay lambda in MCTS value estimation($\lambda$) | 0.8 |
| Exponential factor in Weighted-Advantage($\alpha$) | 3 |

Table 1: Hyper-parameters for MAZero in SMAC environments

### B.3 DETAILS OF BASELINE ALGORITHMS IMPLEMENTATION

MAMBA (Egorov & Shpilman, 2022) is executed based on the open-source implementation: https://github.com/jbr-ai-labs/mamba with the hyper-parameters in Table 2.

| Parameter | Setting |
|---|---|
| Batch size | 256 |
| GAE $\lambda$ | 0.95 |
| Entropy coefficient | 0.001 |
| Entropy annealing | 0.99998 |
| Number of updates | 4 |
| Epochs per update | 5 |
| Update clipping parameter | 0.2 |
| Actor Learning rate | $5 \times 10^{-4}$ |
| Critic Learning rate | $5 \times 10^{-4}$ |
| Discount factor | 0.99 |
| Model Learning rate | $2 \times 10^{-4}$ |
| Number of epochs | 60 |
| Number of sampled rollouts | 40 |
| Sequence length | 20 |
| Rollout horizon $H$ | 15 |
| Buffer size | $2.5 \times 10^5$ |
| Number of categoricals | 32 |
| Number of classes | 32 |
| KL balancing entropy weight | 0.2 |
| KL balancing cross entropy weight | 0.8 |
| Gradient clipping | 100 |
| Trajectories between updates | 1 |
| Hidden size | 256 |

Table 2: Hyper-parameters for MAMBA in SMAC environments

QMIX (Rashid et al., 2020) is executed based on the open-source implementation: `https://github.com/oxwhirl/pymarl` with the hyper-parameters in Table 3.

| Parameter | Setting |
|---|---|
| Batch size | 32 |
| Buffer size | 5000 |
| Discount factor | 0.99 |
| Actor Learning rate | $5 \times 10^{-4}$ |
| Critic Learning rate | $5 \times 10^{-4}$ |
| Optimizer | RMSProp |
| RMSProp $\alpha$ | 0.99 |
| RMSProp $\epsilon$ | $10^{-5}$ |
| Gradient clipping | 10 |
| $\epsilon$-greedy | $1.0 \rightarrow 0.05$ |
| $\epsilon$ annealing time | 50000 |

Table 3: Hyper-parameters for QMIX in SMAC environments

QPLEX (Wang et al., 2020a) is executed based on the open-source implementation: `https://github.com/wjh720/QPLEX` with the hyper-parameters in Table 4.

| Parameter | Setting |
|---|---|
| Batch size | 32 |
| Buffer size | 5000 |
| Discount factor | 0.99 |
| Actor Learning rate | $5 \times 10^{-4}$ |
| Critic Learning rate | $5 \times 10^{-4}$ |
| Optimizer | RMSProp |
| RMSProp $\alpha$ | 0.99 |
| RMSProp $\epsilon$ | $10^{-5}$ |
| Gradient clipping | 10 |
| $\epsilon$-greedy | $1.0 \rightarrow 0.05$ |
| $\epsilon$ annealing time | 50000 |

Table 4: Hyper-parameters for QPLEX in SMAC environments

RODE (Wang et al., 2020b) is executed based on the open-source implementation: `https://github.com/TonghanWang/RODE` with the hyper-parameters in Table 5.

| Parameter | Setting |
|---|---|
| Batch size | 32 |
| Buffer size | 5000 |
| Discount factor | 0.99 |
| Actor Learning rate | $5 \times 10^{-4}$ |
| Critic Learning rate | $5 \times 10^{-4}$ |
| Optimizer | RMSProp |
| RMSProp $\alpha$ | 0.99 |
| RMSProp $\epsilon$ | $10^{-5}$ |
| Gradient clipping | 10 |
| $\epsilon$-greedy | $1.0 \rightarrow 0.05$ |
| $\epsilon$ annealing time | $50K \sim 500K$ |
| number of clusters | $2 \sim 5$ |

Table 5: Hyper-parameters for RODE in SMAC environments

CDS (Li et al., 2021) is executed based on the open-source implementation: `https://github.com/lich14/CDS` with the hyper-parameters in Table 6.

| Parameter | Setting |
|---|---|
| Batch size | 32 |
| Buffer size | 5000 |
| Discount factor | 0.99 |
| Actor Learning rate | $5 \times 10^{-4}$ |
| Critic Learning rate | $5 \times 10^{-4}$ |
| Optimizer | RMSProp |
| RMSProp $\alpha$ | 0.99 |
| RMSProp $\epsilon$ | $10^{-5}$ |
| Gradient clipping | 10 |
| $\epsilon$-greedy | $1.0 \to 0.05$ |
| $\beta$ | 0.05 |
| $\beta_1$ | 0.5 |
| $\beta_2$ | 0.5 |
| $\lambda$ | 0.1 |
| attention regulation coefficient | $10^{-3}$ |

Table 6: Hyper-parameters for CDS in SMAC environments

MAPPO (Yu et al., 2022) is executed based on the open-source implementation: `https://github.com/marlbenchmark/on-policy` with the hyper-parameters in Table 7.

| Parameter | Setting |
|---|---|
| Recurrent data chunk length | 10 |
| Gradient clipping | 10 |
| GAE $\lambda$ | 0.95 |
| Discount factor | 0.99 |
| Value loss | huber loss |
| Huber delta | 10.0 |
| Batch size | num envs × buffer length × num agents |
| Mini batch size | batch size / mini-batch |
| Optimizer | Adam |
| Optimizer: learning rate | $5 \times 10^{-4}$ |
| Optimizer: RMSprop epsilon | $10^{-5}$ |
| Optimizer: weight decay | 0 |

Table 7: Hyper-parameters for MAPPO in SMAC environments

## B.4 DETAILS OF THE BANDIT EXPERIMENT

The bandit experiment showed in Figure 1 compares the behavior cloning loss and the AWPO loss on a 100-armed bandit, with action values $0, \cdots, 99$ and sampling time $B = 2$. The policy $\pi^{\text{BC}}$ and $\pi^{\text{AWPO}}$ are parameterized by Softmax policy, which means

$$\pi^{\text{BC}}(a; \theta^{\text{BC}}) = \frac{\exp(\theta_a^{\text{BC}})}{\sum_b \exp(\theta_b^{\text{BC}})}$$

$$\pi^{\text{AWPO}}(a; \theta^{\text{AWPO}}) = \frac{\exp(\theta_a^{\text{AWPO}})}{\sum_b \exp(\theta_b^{\text{AWPO}})}$$

where $\theta^{\text{BC}}, \theta^{\text{AWPO}} \in \mathbb{R}^{100}$ are randomly initialized and are identical in the beginning.

To exclude the influence of stochasticity in the search algorithm and facilitate a more precise and fair comparison of the differences in loss functions, we made three targeted adjustments.

1. Let the subset of sampled action be $T$, we denote the target policy $\omega(a) = \mathbb{I}(a = \arg\max_{b \in T} \text{value}(b))$.
2. We calculate the expectation of the loss over the randomness of all possible $T$s.
3. We normalize the advantage in AWPO into 0-mean-1-std and choose $\alpha = 1$.

Formally, we have

$$l^{\text{BC}}(\theta) = -\sum \log \pi_\theta(a)\overline{t_\theta(a)}$$

$$l^{\text{BC}}(\theta) = -\sum \log \pi_\theta(a)\overline{t_\theta(a)} \exp\left(\text{adv}(a)\right)$$

where $t(a) = \left(\sum_{b \geq a} \pi_\theta(b)\right)^K - \left(\sum_{b > a} \pi_\theta(b)\right)^K$ is the expectation of $\omega(a)$, the over line stands for `stop-gradient`, $\text{adv}(a) = \left(a - \frac{99}{2}\right)\sqrt{\frac{3}{2525}}$ is the normalized advantage.

## C  STANDARDISED PERFORMANCE EVALUATION PROTOCOL

We report our experiments based on the standardised performance evaluation protocol (Agarwal et al., 2021).

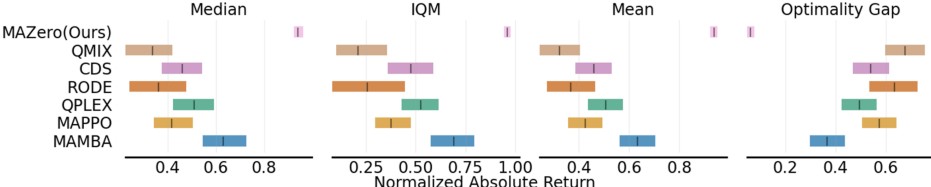

Figure 7: Aggregate metrics on the SMAC benchmark with 95% stratified bootstrap confidence intervals. Higher median, interquartile mean (IQM), and mean, but lower optimality gap indicate better performance.

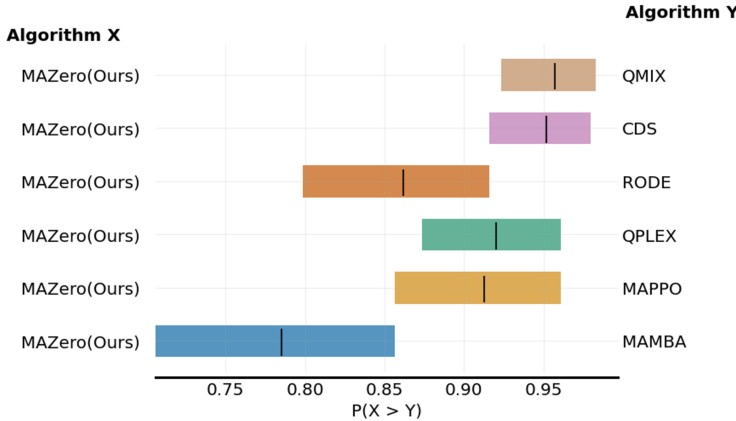

Figure 8: Probabilities of improvement, i.e. how likely it is for MAZero to outperform baselines on the SMAC benchmark.

## D  MORE ABLATIONS

In the experiment section, we list some ablation studies to prove the effectiveness of each component in MAZero. In this section, we will display more results for the ablation study about hyperparameters.

First, we perform an ablation study on the training stage optimizers, contrasting SGD and Adam. The SGD optimizer is prevalent in most MuZero-based algorithms (Schrittwieser et al., 2020; Ye et al., 2021), while Adam is frequently used in MARL environments (Rashid et al., 2020; Wang et al., 2020a; Yu et al., 2022). Figure 9 indicates that Adam demonstrates superior performance and more consistent stability compared to SGD.

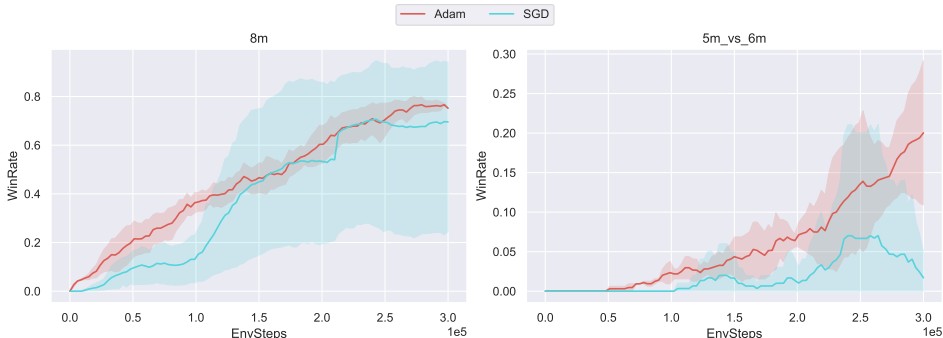

Figure 9: Ablation on optimizer

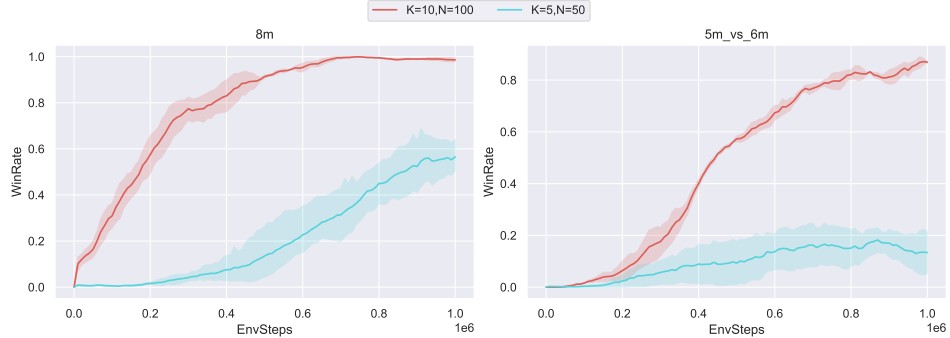

Figure 10: Ablation on sampled scale

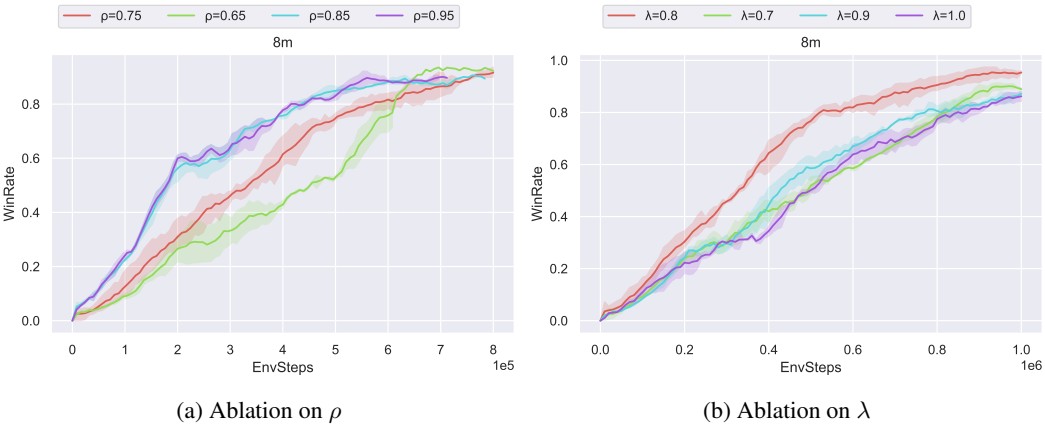

(a) Ablation on $\rho$         (b) Ablation on $\lambda$

Figure 11: Ablation for Hyper-parameters of OS($\lambda$)

In addition, we perform an ablation study on the sampled scale (the action sampling times $K$ and simulation numbers $N$ in MCTS), which has been shown to be essential for final performance in Sampled MuZero (Hubert et al., 2021). Given the limitations of time and computational resources, we only compare two cases: $K = 5, N = 50$ and $K = 10, N = 100$ for easy map *8m* and hard

map *5m_vs_6m*. Figure 10 reveals that our method yields better performance with a larger scale of samples.

We also test the impact of $\rho$ and $\lambda$ in our Optimistic Search Lambda (OS($\lambda$)) algorithm on map *8m*. We test $\rho = 0.65, 0.75, 0.85, 0.95$ by fixing $\lambda = 0.8$, and test $\lambda = 0.7, 0.8, 0.9, 1.0$ by fixing $\rho = 0.75$ for MAZero. Figure 11 shows that the optimistic approach stably improves the sample efficiency, and the $\lambda$ term is useful when dealing with the model error.

We perform an ablation study about MCTS planning in the evaluation stage. The MAZero algorithm is designed under the CTDE framework, which means the global reward allows the agents to learn and optimize their policies collectively during centralized training. The predicted global reward is used in MCTS planning to search for a better policy based on the network prior. Table 8 shows that agents maintain comparable performance without MCTS planning during evaluation, i.e., directly using the final model and local observations without global reward, communication and MCTS planning.

| Map | Env steps | w MCTS | w/o MCTS | performance ratio |
|---|---|---|---|---|
| 3m | 50k | 0.985 ± 0.015 | 0.936 ± 0.107 | 95.0 ± 10.8% |
| 2m_vs_1z | 50k | 1.0 ± 0.0 | 1.0 ± 0.0 | 100 ± 0.0% |
| so_many_baneling | 50k | 0.959 ± 0.023 | 0.938 ± 0.045 | 97.8 ± 4.7% |
| 2s_vs_1sc | 100k | 0.948 ± 0.072 | 0.623 ± 0.185 | 65.7 ± 19.5% |
| 2c_vs_64zg | 400k | 0.893 ± 0.114 | 0.768 ± 0.182 | 86.0 ± 20.4% |
| 5m_vs_6m | 1M | 0.875 ± 0.031 | 0.821 ± 0.165 | 93.8 ± 18.9% |
| 8m_vs_9m | 1M | 0.906 ± 0.092 | 0.855 ± 0.127 | 94.4 ± 14.0% |
| 10m_vs_11m | 1M | 0.922 ± 0.064 | 0.863 ± 0.023 | 93.6 ± 2.5% |
| average performance | | 100% | 90.1 ± 11.3% | |

Table 8: Ablation for using MCTS during evaluation in SMAC environments

# E    EXPERIMENTS ON SINGLE-AGENT ENVIRONMENTS

We have considered demonstrating the effectiveness of OS($\lambda$) and AWPO in single-agent decision problems with large action spaces. This might help establish the general applicability of these techniques beyond the multi-agent SMAC benchmark.

We choose the classical LunarLander environment as the single-agent benchmark, but discretize the 2-dimensional continuous action space into 400 discrete actions. Additionally, we select the Walker2D scenario in MuJoCo environment and discretize each dimension of continuous action space into 7 discrete actions, i.e., $6^7 \approx 280,000$ legal actions. Tables 9 and 10 illustrates the results where both techniques greatly impact learning efficiency and final performance.

| Environment | Env steps | MAZero | w/o OS($\lambda$) and AWPO |
|---|---|---|---|
| | 250k | **184.6±22.8** | 104.0±87.5 |
| LunarLander | 500k | **259.8±12.9** | 227.7±56.3 |
| | 1M | **276.9±2.9** | 274.1±3.5 |

Table 9: Ablation for using OS($\lambda$) and AWPO in LunarLander environments

| Environment | Env steps | MAZero | w/o OS($\lambda$) and AWPO | TD3 | SAC |
|---|---|---|---|---|---|
| | 300k | **3424 ± 246** | 2302 ± 472 | 1101 ± 386 | 1989 ± 500 |
| Walker2D | 500k | **4507 ± 411** | 3859 ± 424 | 2878 ± 343 | 3381 ± 329 |
| | 1M | **5189 ± 382** | 4266 ± 509 | 3946 ± 292 | 4314 ± 256 |

Table 10: Ablation for using OS($\lambda$) and AWPO in Walker2D environments

## F  EXPERIMENTS ON OTHER MULTI-AGENT ENVIRONMENTS

It is beneficial to validate the performance of MAZero in other tasks beyond the SMAC benchmark. We further benchmark MAZero on Google Research Football(GRF) (Kurach et al., 2020), *academy_pass_and_shoot_with_keeper* scenario and compare our methods with several model-free baseline algorithms. Table 11 shows that our method outperforms baselines in terms of sample efficiency.

| Environment | Env steps | MAZero | CDS | RODE | QMIX |
|---|---|---|---|---|---|
| academy_pass_and _shoot_with_keeper | 500k | **0.123 ± 0.089** | 0.069 ± 0.041 | 0 | 0 |
| | 1M | **0.214 ± 0.072** | 0.148 ± 0.117 | 0 | 0 |
| | 2M | **0.619 ± 0.114** | 0.426 ± 0.083 | 0.290 ± 0.104 | 0 |

Table 11: Comparisons against baselines in GRF.

## G  DERIVATION OF AWPO LOSS

To prove that AWPO loss (Equation (10)) is essentially solving the corresponding constrained optimization problem (Equation (11)), we only need to prove the closed form of Equation (11) is Equation (12).

We follow the derivation in Nair et al. (2021). Note that the following optimization problem

$$
\eta^* = \arg\max_{\eta} \mathbb{E}_{\mathbf{a} \sim \pi(\cdot|\mathbf{s})} \left[ A(\mathbf{s}, \mathbf{a}) \right]
$$
$$
\text{s.t. } \mathrm{KL}\left( \eta(\cdot \mid \mathbf{s}) \| \pi(\cdot \mid \mathbf{s}) \right) \leq \epsilon
$$
$$
\int_{\mathbf{a}} \eta(\mathbf{a} \mid \mathbf{s}) \mathrm{d}\mathbf{a} = 1. \tag{15}
$$

has Lagrangian

$$
\mathcal{L}(\eta, \lambda, \alpha) = \mathbb{E}_{\mathbf{a} \sim \eta(\cdot|\mathbf{s})} \left[ A(\mathbf{s}, \mathbf{a}) \right]
$$
$$
+ \lambda \left( \epsilon - D_{\mathrm{KL}}\left( \eta(\cdot \mid \mathbf{s}) \| \pi(\cdot \mid \mathbf{s}) \right) \right)
$$
$$
+ \alpha \left( 1 - \int_{\mathbf{a}} \eta(\mathbf{a} \mid \mathbf{s}) d\mathbf{a} \right). \tag{16}
$$

Applying KKT condition, we have

$$
\frac{\partial \mathcal{L}}{\partial \eta} = A(\mathbf{s}, \mathbf{a}) + \lambda \log \pi(\mathbf{a} \mid \mathbf{s}) - \lambda \log \eta(\mathbf{a} \mid \mathbf{s}) + \lambda - \alpha = 0 \tag{17}
$$

Solving the above equation gives

$$
\eta^*(\mathbf{a} \mid \mathbf{s}) = \frac{1}{Z(\mathbf{s})} \pi(\mathbf{a} \mid \mathbf{s}) \exp\left( \frac{1}{\lambda} A(\mathbf{s}, \mathbf{a}) \right) \tag{18}
$$

where $Z(s)$ is a normalizing factor.

To make the KKT condition hold, we can let $\eta > 0$ and use the LICQ (Linear independence constraint qualification) condition.

Plugging the original problem (Equation (11)) into Equation (15) completes our proof.

