# OpenReview forum: "Efficient Multi-agent Reinforcement Learning by Planning"
_ICLR.cc/2024/Conference — ICLR 2024 poster_

### Official Review · Reviewer_efig · 2023-10-30

**Soundness:** 3 good
**Presentation:** 3 good
**Contribution:** 3 good
**Rating:** 6
**Confidence:** 5

**Summary:**

This work proposes a model-based multi-agent RL approach similar to MuZero, performing MCTS over the joint action space of all agents as imagined by learned models of the dynamics and reward. The authors propose to use optimism (Optimistic Search Lambda in conjunction with Advantage-Weighted Policy Optimization) to improve the performance over the behavioral cloning loss used in Sampled MuZero. Experiments on the SMAC benchmark compare the performance of this method with some previously proposed model-based and model-free MARL methods.

**Strengths:**

The paper demonstrates better or similar performance compared with several baselines. The paper is easy to follow (at least for someone familiar with MuZero and related works). Regarding novelty, I have not found any previous application of MuZero to multi-agent setting.

**Weaknesses:**

There are several major weaknesses of the paper.

1. The empirical validation of the method is very weak.
    1. Despite the authors proposing the method to tackle large action spaces and mentioning the 27m_vs_30m settings as a motivating example in Section 3.2, there is no experiment in these large action space settings.
    2. There’s a lack of experimental validation beyond the SMAC benchmark, and the same benchmarks settings are used across all results and ablations. For example, Google Research Football and Multi-agent MuJoCo would be candidates for other tasks.
    3. The SMAC benchmark is outdated and should be replaced by the SMACv2 benchmark, as many tasks have been shown to be trivially solvable due to the lack of stochasticity in the SMAC benchmark.
    4. The baselines used are relatively few compared to other published MARL works. There should be comparison with other recent methods like MBVD, RODE, CDS, etc.
    5. It’s not clear how many seeds the work used for the main Table 3. The work mentions 3 seeds for followup ablations, but this is too few to ensure that the performance is not due to luck. The results should be reported across at least 10 seeds.
2. To my knowledge, the search-based component of methods like MuZero are very reduced and not significantly useful for many non-board game tasks. For example, while MuZero utilizes a large search depth for Go, it uses a very small search depth for Atari, which raises the questions of how much performance gain the search component contributes. This work mentions that it uses 5 unroll steps, which is still very small. For MCTS-based works like this one, there should be empirical investigations with respect to the number of unroll steps at both training and test time to see whether search is offering any benefits.
4. As this work mostly treats multi-agent decision problem as a single-agent RL with a very large action space, the authors should demonstrate $OS(\lambda)$ and AWPO more extensively on (single-agent) decision problems with large action space; indeed, the author could even solely consider these single-agent settings rather than bringing in the multi-agent SMAC benchmark at all. The single-agent bandit experiment in Figure 1 is too simplistic and insufficient to demonstrate the authors’ point. Also, in very large action spaces, MCTS with a limited budget (e.g. $N = 100$ MCTS simulations as considered in this work) seems very sparse and unlikely to simulate the same action multiple times from a given state. This would significantly weaken the motivation for $OS(\lambda)$, which relies on simulating each action many times from a given state. In such large action spaces, a very relevant heuristics baseline is to sample $N$ simulations (each starting with a different action) with the learned policy and simply follow the action given by the best simulation. Overall, there's insufficient experiments demonstrating that $OS(\lambda)$ and AWPO solve the stated large action space problem without suffering from unintended side-effects of the additional optimism, and there should be more trivial heuristics baselines which may also work well for searching large action spaces with a fixed budget.

**Questions:**

Is MCTS being performed at evaluation time, or is only the learned policy used? If search is being performed at evaluation time, there should be a quantification of the search overhead compared with baselines.

---

> ### Author Response · Authors · 2023-11-22
>
> We would like to express our sincere gratitude for your valuable time, constructive comments, and dedication to helping improve the quality of our paper. The insightful feedback have been instrumental in identifying areas for improvement and refining our presentation. In response to your questions and concerns, we have made the following revisions and clarifications:
>
> **Q1: The search-based component of methods like MuZero are reduced and not significantly useful for many non-board game tasks.**
>
> Thank you for raising concerns about the usefulness of search-based methods like MuZero in non-board game tasks. While it is true that model-free RL has achieved remarkable success in various tasks, it is important to recognize the growing importance of search-based methods, such as MCTS, in recent years.
>
> Although search-based methods may seem less significant compared to model-free RL, they  have actually demonstrated valuable contributions across different domains. EfficientZero [1], for example, has achieved impressive results on standard RL benchmarks. According to the paper, it achieves 194.3% mean human performance on the Atari 100k benchmark with only two hours of real-time game experience. Additionally, there are extensions of MuZero in continuous control [2,3] that outperform model-free baselines in terms of data efficiency. These results strongly indicate the effectiveness of search-based methods in these benchmarks.
>
> Moreover, search-based methods have made significant breakthroughs in application domains. For instance, AlphaTensor [4] discovered faster matrix multiplication algorithms, marking the first improvement in this area in 50 years. AlphaDev [5], its sister work, has also found faster sorting algorithms, some of which have been integrated into the standard sort function in the LLVM standard C++ library. Furthermore, these methods have been extended to NP-hard math problems, further demonstrating their potential and applicability.
>
> In the field of AGI research, search-based methods show promise in providing reasonability for LLM-agents. Many recent works [6-8] are exploring the use of search-based methods to enhance the capabilities of large language models, indicating their potential for advancing natural language understanding and reasoning.
>
> Overall, it is crucial to acknowledge that search-based methods have the potential to be utilized in numerous important domains. With this in mind, the objective of this paper is to contribute to the application of search-based methods in the multi-agent domain, serving as a foundation for further exploration and progress in this field.
>
> [1] Ye, Weirui, et al. "Mastering atari games with limited data." *Advances in Neural Information Processing Systems* 34 (2021): 25476-25488.
>
> [2] Hansen, Nicklas, Xiaolong Wang and Hao Su. “Temporal Difference Learning for Model Predictive Control.” *International Conference on Machine Learning* (2022).
>
> [3] Hubert, Thomas, et al. "Learning and planning in complex action spaces." *International Conference on Machine Learning*. PMLR, 2021.
>
> [4] Fawzi, Alhussein, et al. "Discovering faster matrix multiplication algorithms with reinforcement learning." *Nature* 610 (2022): 47-53.
>
> [5] Mankowitz, Daniel J., et al. "Faster sorting algorithms discovered using deep reinforcement learning." *Nature* 618.7964 (2023): 257-263.
>
> [6] Yao, Shunyu, et al. "Tree of Thoughts: Deliberate Problem Solving with Large Language Models." arXiv preprint arXiv:2305.10601 (2023).
>
> [7] Feng, Xidong, et al. "Alphazero-like Tree-Search can Guide Large Language Model Decoding and Training." arXiv preprint arXiv:2309.17179 (2023).
>
> [8] Zhao, Zirui, Wee Sun Lee, and David Hsu. "Large Language Models as Commonsense Knowledge for Large-Scale Task Planning." arXiv preprint arXiv:2305.14078 (2023).
>
> **Q2: For example, while MuZero utilizes a large search depth for Go, it uses a very small search depth for Atari, which raises the questions of how much performance gain the search component contributes. This work mentions that it uses 5 unroll steps, which is still very small.**
>
> It appears there may be a misconception about its role in our model. The ”*unroll step*” refers to the expansion steps during the model-learning phase and is distinct from the search depth used in the search phase. In our work, we set the *unroll step* to 5, mirroring the configuration used by MuZero in board games like Go. This setting, while seemingly modest, allows us to maintain a balance between learning a robust model and computational efficiency. It's important to note that despite an *unroll step* of 5, our model can achieve a search depth exceeding 20, with a total number of simulations ($N$) being 50. This demonstrates that a smaller *unroll step* is effective and sufficient for learning an accurate model, as also evidenced by MuZero's performance in complex games. We chose this approach primarily for efficiency while ensuring minimal model error.

---

> ### Author Response · Authors · 2023-11-22
>
> **Q3: There is no experiment in large action space settings like *27_vs_30m*.**
>
> We are sorry for mentioning *27m_vs_30m* in our motivation section while remaining it unevaluated in our experiment section due to the relatively large time consumption for our algorithm in this map (MAPPO uses 10M data for this map). Because of the stress of computing power, we cannot provide another experiment for it. However, we would like to point out that there are still similar tasks like *10m_vs_11m* whose action space is $17^{10}\approx 2\times 10^{12}$, which is also considerably larger than the most complex environment evaluated by Sampled Muzero (Go, $361$ actions). Therefore, while we couldn't include *27m_vs_30m*, our chosen experiments still showcase the algorithm's robustness in complex environments.
>
> We hope this clarification assures you of our algorithm's applicability to a range of scenarios, including those with large action spaces, even though not all could be directly tested due to resource constraints.
>
> **Q4: There’s a lack of experimental validation beyond the SMAC benchmark, and the same benchmarks settings are used across all results and ablations. For example, Google Research Football and Multi-agent MuJoCo would be candidates for other tasks.**
>
> We agree on the importance of validating the algorithm's performance in other tasks beyond the SMAC benchmark. It would be beneficial to validate the performance of MAZero in other tasks beyond the SMAC benchmark. We further benchmark MAZero on Google Research Football *academy_pass_and_shoot_with_keeper*, the result is shown below(see Appendix E, Table 11), where our method outperforms baselines in terms of sample efficiency.
>
> - Comparisons on *academy_pass_and_shoot_with_keeper,* Google Research Football.
>
>
>     | Env steps | MAZero | CDS | RODE | QMIX |
>     | --- | --- | --- | --- | --- |
>     | 500k | **0.123 ± 0.089** | 0.069 ± 0.041 | 0 | 0 |
>     | 1M | **0.214 ± 0.072** | 0.148 ± 0.117 | 0 | 0 |
>     | 2M | **0.619 ± 0.114** | 0.426 ± 0.083 | 0.290 ± 0.104 | 0 |
>
> **Q5: The SMAC benchmark is outdated and should be replaced by the SMACv2 benchmark, as many tasks have been shown to be trivially solvable due to the lack of stochasticity in the SMAC benchmark.**
>
> We appreciate the reviewer for the suggestion. Nevertheless, we did not evaluate MAZero in SMACv2 in our paper for three reasons:
>
> - **Focus on Deterministic Environments**: The primary focus of our research is deterministic environments, as opposed to stochastic ones. MAZero, building upon the legacy of MuZero, is designed for deterministic models. Our Optimistic Search Lambda technique, in particular, is tailored to leverage deterministic models. While testing on SMACv2 is possible, we believe it might detract from the clarity of our research objectives, which are centered on deterministic settings.
> - **Baseline Comparability**: The SMAC benchmark has been widely used for evaluating numerous baseline algorithms. Some baselines like MAMBA are benchmarked on SMAC, not SMACv2. To facilitate straightforward reproduction of results and ensure fair and relevant comparisons, we opted for the SMAC benchmark. This choice ensures that our findings are directly comparable to a broad range of existing research.
> - **Relevance of SMAC Despite Low Stochasticity**: Even though SMAC is recognized for its lower stochasticity compared to SMACv2, it still presents a meaningful and non-trivial challenge for multi-agent model learning and searching. Evaluating MAZero on SMAC effectively demonstrates our model's capabilities and the efficacy of our search techniques within this context.
>
> We acknowledge the evolving landscape of benchmarks in this field and the potential value of including SMACv2 in future work. However, for the purposes of this paper, we believe that SMAC remains a relevant and suitable choice for demonstrating the strengths and applications of MAZero.
>
> **Q6: The baselines used are relatively few compared to other published MARL works. There should be comparison with other recent methods like MBVD, RODE, CDS, etc.**
>
> We appreciate the reviewer’s suggestion. We have added RODE and CDS as our baseline (see Figure 3) and results show that MAZero significantly outperform them, especially in terms of sample efficiency. We do not add MBVD as the baseline since we fail to find any open-source implementation of it.
>
> **Q7: It’s not clear how many seeds the work used for the main Figure 3. The work mentions 3 seeds for followup ablations, but this is too few to ensure that the performance is not due to luck. The results should be reported across at least 10 seeds.**
>
> We have considered the importance of using more seeds for the main Figure 3 to ensure robust evaluation results. We have conducted extra experiments and update Figure 3 with 10 random seeds.

---

> ### Author Response · Authors · 2023-11-22
>
> **Q8: The authors should demonstrate OS($\lambda$) and AWPO more extensively on (single-agent) decision problems with large action space**
>
> We have considered demonstrating the effectiveness of OS($\lambda$) and AWPO in single-agent decision problems with large action spaces. Specifically, we choose *LunarLander* environment but discretize action space into $400$ actions. Additionally, we select the *Walker2D* scenario in MoJoCo environment and discretize each dimension of continuous action space into 7 actions, i.e., $6^7 \approx 280,000$ legal actions. Tables 9 and 10 in Appendix D present the results, demonstrating the enhancement of both techniques in terms of learning efficiency and final performance.
>
> - Ablation study on *LunarLander*.
>
>
>     | Env steps | MAZero | w/o OS($\lambda$) and AWPO |
>     | --- | --- | --- |
>     | 250k | **184.6 ± 22.8** | 104.0 ± 87.5 |
>     | 500k | **259.8 ± 12.9** | 227.7 ± 56.3 |
>     | 1M | **276.9 ± 2.9** | 274.1 ± 3.5 |
> - Ablation study on *Walker2D*, MuJoCo.
>
>
>     | Env steps | MAZero | w/o OS($\lambda$) and AWPO | TD3 | SAC |
>     | --- | --- | --- | --- | --- |
>     | 300k | **3424 ± 246** | 2302 ± 472 | 1101 ± 386 | 1989 ± 500 |
>     | 500k | **4507 ± 411** | 3859 ± 424 | 2878 ± 343 | 3381 ± 329 |
>     | 1M | **5189 ± 382** | 4266 ± 509 | 3946 ± 292 | 4314 ± 256 |
>
> **Q9: In very large action spaces, MCTS with a limited budget (e.g. $N=100$ MCTS simulations as considered in this work) seems very sparse and unlikely to simulate the same action multiple times from a given state. This would significantly weaken the motivation for OS($\lambda$), which relies on simulating each action many times from a given state.**
>
> Thank you for highlighting concerns regarding the application of OS($\lambda$) in large action spaces with limited MCTS simulation budgets. We recognize the importance of clarity in explaining our methods and appreciate the opportunity to elucidate further.
>
> OS($\lambda$), as employed in our work, is built upon the framework of Sampled MCTS, wherein each node expansion is conducted exactly once. This is a crucial aspect that seems to have been misunderstood by the reviewer. When a node $u$ is expanded, its set of child nodes, typically limited to a size of $K$ (about $5$ or $10$ in our settings), is determined through an on-policy distribution $\beta$. This 'sampled' aspect ensures that each expanded node receives a unique value estimate through a forward pass of the value network, followed by backpropagation to the root. Later, when an expanded node is revisited, we recursively choose a son of it and do the same thing. Therefore, each expanded node will get its value network forwarded exactly once.
>
> Contrary to the classical sampled MCTS, the final value of a node is the average of all estimations in its subtree (each node $x$ in its subtree constitutes an estimation in the form of $r_1+r_2+…+v(x)$), which is too pessimistic when the model is deterministic and action space is large. This motivates us to design a more optimistic value of expanded nodes.
>
> When action space is large and the simulation budget $N$ is limited, the sampling procedure in classical sampled MCTS restricts the degree of the tree to at most $K$ ($K \ll N$). This restriction enables deeper search depth, mitigating the sparsity issues. While OS($\lambda$) performs optimistic value estimation, enhancing the search efficiency by effectively navigating the limited tree structure.
>
> We hope this clarification underscores the suitability and efficacy of OS($\lambda$) in handling the challenges posed by large action spaces and limited simulation budgets.
>
> **Q10: Is MCTS being performed at evaluation time, or is only the learned policy used? If search is being performed at evaluation time, there should be a quantification of the search overhead compared with baselines.**
>
> As requested by the reviewer, we have conducted additional ablation study about whether or not perform MCTS in Appendix C, Table 8. Results on SMAC environments show that agents maintain comparable performance without MCTS planning during evaluation.
>
> | Map | Env steps | w MCTS | w/o MCTS | performance ratio |
> | --- | --- | --- | --- | --- |
> | 3m | 50k | 0.985 ± 0.015 | 0.936 ± 0.107 | 95.0 ± 10.8% |
> | 2m_vs_1z | 50k | 1.0 ± 0.0 | 1.0 ± 0.0 | 100 ± 0.0% |
> | so_many_baneling | 50k | 0.959 ± 0.023 | 0.938 ± 0.045 | 97.8 ± 4.7% |
> | 2s_vs_1sc | 100k | 0.948 ± 0.072 | 0.623 ± 0.185 | 65.7 ± 19.5% |
> | 2c_vs_64zg | 400k | 0.893 ± 0.114 | 0.768 ± 0.182 | 86.0 ± 20.4% |
> | 5m_vs_6m | 1M | 0.875 ± 0.031 | 0.821 ± 0.165 | 93.8 ± 18.9% |
> | 8m_vs_9m | 1M | 0.906 ± 0.092 | 0.855 ± 0.127 | 94.4 ± 14.0% |
> | 10m_vs_11m | 1M | 0.922 ± 0.064 | 0.863 ± 0.023 | 93.6 ± 2.5% |
> | average performance |  | 100% | 90.1 ± 11.3 % |  |

---

> > ### Comment · Reviewer_efig · 2023-11-22
> >
> > I thank the authors for addressing several of my concerns. Here I list my updated thoughts about the paper:
> > 1. After revisiting the literature on MuZero and EfficientZero, I do agree with the authors that MCTS-based approaches (in the representation space) may be useful for model-free settings like Atari (and SMAC, as shown in this paper). I am well familiar with the effectiveness of AlphaZero/MCTS for naturally model-based settings like Go, matrix multiplication (AlphaTensor), and assembly code generation. This is also demonstrated by the author's Appendix C Table 8: searching in the representation space does help.
> > 2. I am more encouraged by the author's additional comparisons with Sampled MuZero (i.e. MAZero w/o OS($\lambda$) and AWPO) in LunarLander and Walker2D. However, selecting one Google Research Football task is insufficient in eliminating doubts of cherrypicking, and I would like to see more tasks.
> > 3. As helped by the author's clarifications, I can see why the authors have chosen relatively small SMAC environments now, due to the inherent scalability limitations of Sampled MuZero, despite the addition of proposed OS($\lambda$) and AWPO. I think these choices of environments is fairly reasonable.
> >
> > I would be happy to raise my score to a 5 for now. Reflecting the above thoughts, my main outstanding doubts are:
> > 1. Given the algorithm's strong dependence on the performance of Sampled MuZero (as noted in my point #3 above), why is Sampled MuZero *not* a part of Figure 3 but instead relegated to ablation in Figure 5? I'm not thoroughly convinced that the method consistently outperforms Sampled MuZero, given only two SMAC comparisons (8m and 2c_vs_64zg) as well as LunarLander and Walker2D, with the majority of SMAC environments left out. **Why was Sampled MuZero not tested for 5m_vs_6m, 8m_vs_9m, 10m_vs_11m, 3m, 2m_vs_1z, so_many_banelings, and 2s_vs_1sc, while the 8m environment was not even introduced in Table 3?**
> > 2. A single Google Research Football task does not adequately demonstrate the performance of the method in that setting. For example, CDS uses three tasks: academy_3_vs_1_with_keeper, academy_counterattack_hard, and a designed full-field scenario 3_vs_1_with_keeper (full field). I would ideally want to see both MAZero and Sampled MuZero on multiple GRF tasks.
> >
> > Unfortunately, I know the authors have limited time to respond to these concerns. Though I'm keeping my score at a 5 for now, I'd be happy to keep an open mind if discussions with other reviewers are needed to make a decision.

---

> ### Author Response · Authors · 2023-11-23
> **Thank you for the feedback**
>
> Thank you for raising the score and acknowledging our rebuttal results. We greatly appreciate your positive response to our additional experiments. Regarding your question about conducting ablation on only two SMAC maps, we specifically chose those maps because they exhibited the most apparent differences. This selection allowed us to effectively highlight the effects of the MCTS improvements. By focusing on these maps, we were able to demonstrate the impact of the ablation more clearly. Furthermore, we would like to clarify that the results presented in Table 3 were evaluated directly using the final model trained with MAZero, rather than starting from scratch. This approach enabled us to compare the performance whether or not conduct MCTS during evaluation in a more practical setting. Furthermore, we have extended the comparison between MAZero and Sampled MuZero across all the maps tested. Please note that in the supplemental experiment, Sampled MuZero uses the shared dynamic network we proposed; otherwise, the performance is very poor in SMAC.
>
> |  | steps | MAZero | Sampled MuZero(MAZero network) | Sampled MuZero(flattened network) |
> | --- | --- | --- | --- | --- |
> | 3m | 50k | 0.985 ± 0.015 | 0.951 ± 0.034 | 0.150 ± 0.178 |
> | 2m_vs_1z | 50k | 1.0 ± 0.0 | 1.0 ± 0.0 | 0.803 ± 0.091 |
> | so_many_baneling | 50k | 0.959 ± 0.023 | 0.747 ± 0.112 | 0.0 ± 0.0 |
> | 2s_vs_1sc | 100k | 0.948 ± 0.072 | 0.721 ± 0.124 | 0.0 ± 0.0 |
> | 2c_vs_64zg | 400k | 0.893 ± 0.114 | 0.374 ± 0.124 | 0.0 ± 0.0 |
> | 8m | 1M | 0.993 ± 0.009 | 0.751 ± 0.078 | 0.0 ± 0.0 |
> | 5m_vs_6m | 1M | 0.875 ± 0.031 | 0.696 ± 0.047 | 0.0 ± 0.0 |
> | 8m_vs_9m | 1M | 0.906 ± 0.092 | 0.812 ± 0.089 | 0.0 ± 0.0 |
> | 10m_vs_11m | 1M | 0.922 ± 0.064 | 0.738 ± 0.205 | 0.0 ± 0.0 |
>
> We understand your concerns regarding the limited number of Google Research Football tasks we evaluated. Due to time constraints, we were unable to test our method on multiple tasks. However, we believe that the results we have presented lay a solid foundation for further investigation. We greatly appreciate your open-mindedness and willingness to consider discussions with other reviewers if necessary. We hope that our future work will address these concerns and provide more comprehensive results across a wider range of tasks.

---

> > ### Comment · Reviewer_efig · 2023-11-23
> >
> > Thank you for providing these additional results. These have addressed my previous concern that the proposed OS($\lambda$) and AWPO might not provide much benefit over Sampled MuZero. I trust that the authors will incorporate all discussed results during the rebuttal into the paper, preferably incorporating Sampled MuZero (or MAZero w/o OS($\lambda$) and AWPO) into Figure 3 as full curves rather than just the win rate at the end (and perhaps removing Figure 5).
> >
> > I will raise my score to a 6. My concerns about the predominantly using the SMAC environments remain.

---

### Official Review · Reviewer_MHMy · 2023-10-31

**Soundness:** 2 fair
**Presentation:** 2 fair
**Contribution:** 2 fair
**Rating:** 6
**Confidence:** 4

**Summary:**

The authors introduce a novel algorithm, MAZero, based on the model-based ideas in MuZero, for the multi agent setting in order to improve problems with sample efficiency. Because the model based approach requires roll-outs to select policies, an efficient method of tree search is required, and so Optimistic Search Lambda (a method which weights optimistic values more highly) and Advantage-Weighted Policy Optimization (which uses a novel policy loss function based on the optimistic search lambda values).

The authors then compare this model based MARL algorithm to a number of others on the Starcraft Multi-Agent Challenge followed by a number of ablations of the different methods.

**Strengths:**

The paper is in general well-written, and clear, with thorough appendices for the details.

The results given show that this model-based approach is not only more sample efficient than model-free approaches in the MARL setting but also, importantly, tractable when the Optimistic Search Lambda Algorithm and Weighted Policy Optimization are used within the tree-search. While these results are not surprising, such an approach has not been taken before and so there is definitely originality and significance to these results within the domain explored.

**Weaknesses:**

The major issue comes down to the single, very specific domain that this has been tested on. While the results are, as stated above, impressive, they are only impressive in this single domain, and it would not seem difficult to show that they are just as significant in other domains with different types of action and state spaces (continuous, discrete, visual, tabular).

In addition, I believe that it should become standard within the community to utilise the evaluation protocol of Gorsanne et al (https://arxiv.org/abs/2209.10485) in the Multi-agent setting. These standard practices have not been followed and I believe weaken what could be a strong case for the effectiveness of these algorithms.

No discussion is given to hyper parameter tuning, or the choice of hyperparameters for the baselines that MAZero is being compared against.

On a stylistic note, the paper has a lot of grammatical typos and needs to be gone over thoroughly. An LLM should be able to pick up all of these mistakes.

I would not use the word "ingenious" in the abstract to describe your own algorithm, or, as later used "audacious".
CTDE is not spelled out the first time it is mentioned.
The different loss terms in equation 2 are not spelled out.
Is Figure 1 a single seed? It looks remarkably smooth.

**Questions:**

The questions all relate to the weaknesses described in the previous section.

---

> ### Author Response · Authors · 2023-11-22
>
> Thank you very much for your thoughtful comments, which have allowed us to improve the quality of our manuscript and to post a revised version. In what follows, we address the raised questions and weaknesses point-by-point.
>
> **Q1: Show that they are just as significant in other domains with different types of action and state spaces (continuous, discrete, visual, tabular).**
>
> Thank you for highlighting the broader applicability of our work to various domains with different types of action and state spaces. You've rightly pointed out that our current focus with MAZero has been primarily on discrete environments. In an effort to extend our evaluation to continuous tasks, we employed a method of discretizing the continuous action space into a larger discrete action space. The results of this adaptation can be seen in Tables 9 and 10 of our paper, where MAZero demonstrates notable performance advantages in comparison to baseline algorithms. Due to the time limitation, we do not evaluate MAZero on visual and tabular tasks.
>
> **Q2: Use the evaluation protocol of Gorsanne et al (https://arxiv.org/abs/2209.10485)**
>
> We agree on the importance of utilizing standard evaluation protocol within the community. Because of the stress of computing resources and limitation of time, we cannot restart the whole experiments under such a standard protocol during the response period. We will reorganize the results of all experiments in the next few weeks.
>
> **Q3: No discussion is given to hyper parameter tuning, or the choice of hyper-parameters for the baselines that MAZero is being compared against.**
>
> We have discussed some crucial hyper-parameters (e.g. $\rho, \lambda$ used in OS($\lambda$)) in ablation study in Appendix C. Detailed hyper-parameters are provided in Table 1, which are common settings in Sampled MuZero and EfficientZero.
>
> We have added the choice of hyper-parameters for the baselines in Appendix B.3.
>
> **Q4: Is Figure 1 a single seed? It looks remarkably smooth.**
>
> We appreciate the reviewer’s careful observation! Yes, only a single seed is used, and it is done intentionally. We will briefly explain the motivation here, the full experimental setting, and details were listed in Appendix B.4. The aim of this experiment is to compare the behavior of these two different loss functions, i.e., BC and AWPO. To ensure that the comparison was as controlled and unbiased as possible, we eliminated potential variables that may influence the outcome. In this way, we
>
> - eradicate the randomness in policy evaluation by calculating the expected reward to evaluate the policies.
> - choose the same parameterization (softmax policy), and the same random seed (which only affects the initialization of the policy weights), this is why these two curves have the same starting point.
> - eliminate the randomness in SGD optimization, where the stochasticity comes from the randomness of sampling the subset (cf. Sampled MCTS) by calculating the expectation of the loss (the formula can be found in Appendix B.4), that is why these curves look smooth.
>
> We hope this explanation clarifies the rationale behind our experimental setup and the resulting smoothness observed in Figure 1.

---

> > ### Comment · Reviewer_MHMy · 2023-11-22
> >
> > I thank that authors for their careful responses. I still believe however that the application only to SMAC environments means that the results are hard to extrapolate and thus hard to judge. Using this algorithm on a more diverse set of environments and using robust evaluation metrics would make this a substantially stronger paper.

---

> ### Author Response · Authors · 2023-11-23
>
> Thank you for your quick response. In response to your concerns about the experiments, we have conducted additional experiments beyond the SMAC environments. These experiments include LunarLander, MuJoCo, and Google Research Football. The results presented in Tables 9, 10, and 11 demonstrate the advantages of our algorithm across a diverse range of environments. Furthermore, we have also conducted experiments with an increased seed data. In future versions, we will prioritize considering the evaluation protocol you recommended.
> - Experiments on *LunarLander*.
>
>
>     | Env steps | MAZero | w/o OS($\lambda$) and AWPO |
>     | --- | --- | --- |
>     | 250k | **184.6 ± 22.8** | 104.0 ± 87.5 |
>     | 500k | **259.8 ± 12.9** | 227.7 ± 56.3 |
>     | 1M | **276.9 ± 2.9** | 274.1 ± 3.5 |
> - Experiments on *Walker2D*, MuJoCo.
>
>
>     | Env steps | MAZero | w/o OS($\lambda$) and AWPO | TD3 | SAC |
>     | --- | --- | --- | --- | --- |
>     | 300k | **3424 ± 246** | 2302 ± 472 | 1101 ± 386 | 1989 ± 500 |
>     | 500k | **4507 ± 411** | 3859 ± 424 | 2878 ± 343 | 3381 ± 329 |
>     | 1M | **5189 ± 382** | 4266 ± 509 | 3946 ± 292 | 4314 ± 256 |
> - Experiments on *academy_pass_and_shoot_with_keeper,* Google Research Football.
>
>
>     | Env steps | MAZero | CDS | RODE | QMIX |
>     | --- | --- | --- | --- | --- |
>     | 500k | **0.123 ± 0.089** | 0.069 ± 0.041 | 0 | 0 |
>     | 1M | **0.214 ± 0.072** | 0.148 ± 0.117 | 0 | 0 |
>     | 2M | **0.619 ± 0.114** | 0.426 ± 0.083 | 0.290 ± 0.104 | 0 |

---

> > ### Comment · Reviewer_MHMy · 2023-11-23
> >
> > Thank you for noting this. I have updated my scoring accordingly.

---

### Official Review · Reviewer_6eaK · 2023-11-01

**Soundness:** 3 good
**Presentation:** 3 good
**Contribution:** 3 good
**Rating:** 6
**Confidence:** 3

**Summary:**

The paper presents MAZero, a model-based multi-agent reinforcement learning (MARL) algorithm that combines a centralized model with Monte Carlo Tree Search (MCTS) for policy search. The authors propose an ingenious network structure to facilitate distributed execution and parameter sharing. They introduce two novel techniques, Optimistic Search Lambda (OS(λ)) and Advantage-Weighted Policy Optimization (AWPO), to enhance search efficiency in deterministic environments with sizable action spaces. Extensive experiments on the SMAC benchmark demonstrate that MAZero outperforms model-free approaches in terms of sample efficiency and provides comparable or better performance than existing model-based methods in terms of both sample and computational efficiency.

**Strengths:**

* MAZero is the first empirically effective approach that extends the MuZero paradigm into multi-agent cooperative environments.
* The proposed OS(λ) and AWPO techniques improve search efficiency in large action spaces.
* Extensive experiments on the SMAC benchmark demonstrate the effectiveness of MAZero in terms of sample efficiency and performance.

**Weaknesses:**

* The paper focuses on deterministic environments, and it is unclear how well MAZero would perform in stochastic environments.
* The proposed techniques may not be applicable to all types of multi-agent environments, and further research is needed to generalize the approach.

**Questions:**

* How does MAZero perform in stochastic environments compared to deterministic ones?
* Can the proposed OS(λ) and AWPO techniques be applied to other model-based MARL algorithms?
* Are there any potential drawbacks or limitations of the proposed network structure that the authors have not discussed?

---

> ### Author Response · Authors · 2023-11-22
>
> Thank you very much for your strong support recognizing the novelty of our work. We additionally are grateful for your comments and questions that we aim to address point-by-point in what follows.
>
> **Q1: How does MAZero perform in stochastic environments compared to deterministic ones?**
>
> Our algorithm primarily focuses on deterministic environments, as stated in this paper. We have developed a deterministic model based on the previous work of Muzero. Our OS($\lambda$) technique is specifically designed to optimize this deterministic model. However, we have also conducted additional experiments beyond the SMAC environments, including LunarLander, MuJoCo, and Google Research Football, which exhibit more randomness compared to SMAC. The results in Tables 9, 10, and 11 demonstrate that MAZero performs well to some extent in stochastic environments.
>
> - Ablation study on *LunarLander*.
>
>
>     | Env steps | MAZero | w/o OS($\lambda$) and AWPO |
>     | --- | --- | --- |
>     | 250k | **184.6 ± 22.8** | 104.0 ± 87.5 |
>     | 500k | **259.8 ± 12.9** | 227.7 ± 56.3 |
>     | 1M | **276.9 ± 2.9** | 274.1 ± 3.5 |
> - Ablation study on *Walker2D*, MuJoCo.
>
>
>     | Env steps | MAZero | w/o OS($\lambda$) and AWPO | TD3 | SAC |
>     | --- | --- | --- | --- | --- |
>     | 300k | **3424 ± 246** | 2302 ± 472 | 1101 ± 386 | 1989 ± 500 |
>     | 500k | **4507 ± 411** | 3859 ± 424 | 2878 ± 343 | 3381 ± 329 |
>     | 1M | **5189 ± 382** | 4266 ± 509 | 3946 ± 292 | 4314 ± 256 |
> - Comparisons on *academy_pass_and_shoot_with_keeper,* Google Research Football.
>
>
>     | Env steps | MAZero | CDS | RODE | QMIX |
>     | --- | --- | --- | --- | --- |
>     | 500k | **0.123 ± 0.089** | 0.069 ± 0.041 | 0 | 0 |
>     | 1M | **0.214 ± 0.072** | 0.148 ± 0.117 | 0 | 0 |
>     | 2M | **0.619 ± 0.114** | 0.426 ± 0.083 | 0.290 ± 0.104 | 0 |
>
> **Q2: Can the proposed OS($\lambda$) and AWPO techniques be applied to other model-based MARL algorithms?**
>
> OS($\lambda$) and AWPO are specially designed for MCTS algorithms. We don’t think it can be applied to other model-based MARL algorithms like the dreamer-based MBMARL algorithms (e.g. our baseline method MAMBA).
>
> **Q3: Are there any potential drawbacks or limitations of the proposed network structure that the authors have not discussed?**
>
> We recognize the limitations of the proposed network structure and have outlined potential drawbacks. Some potential drawbacks or limitations of the proposed network structure include:
>
> - The reliance on centralized training, which may not scale well to very large numbers of agents or highly heterogeneous environments.
> - Although MAZero is more sample-efficiency than model-free MARL method, it is relatively computationally expensive.

---

### Official Review · Reviewer_iouX · 2023-11-02

**Soundness:** 3 good
**Presentation:** 3 good
**Contribution:** 4 excellent
**Rating:** 8
**Confidence:** 3

**Summary:**

The paper considers MARL in a game environment, using a model-based approach. The heart of the paper is to extend MuZero, a model-based single agent RL algorithm that incorporates planning, to the multi-agent case in a Dec-POMDP setup.  The new algorithm is called mullti-agent Zero (MAZero). The extension of MuZero is achieved by adding and modifying the 3 functions of MuZero to create 6 functions that incorporate communications and expands how predictions are made.  The paper then focuses on efficient Monte Carlo tree search (MCTS) so that the planning complexity is reasonable. Numerical studies are carried out in Starcraft multi-agent challenge (SMAC) environments.  The experiments compare to model-free and model-based methods as baseline.  The results generally show improved learning efficiency for CDTE execution.

**Strengths:**

The paper is a logical extension of MuZero to the multi-agent case, and builds on those ideas to create six neural network functions that underly the model.  The writing is clear and the various cost functions and parameters are well laid out.  T

The experiments push the MARL problem in terms of action space complexity, and show that the MCTS method is effective in providing good performance with reduced search.  The primary contribution of the paper is to use prediction along with the reduced search in the multi-agent setting.

It appears that the method is generally applicable to Dec-POMDP problems under the assumptions in the paper, in particular the CDTE assumption.

The advantage-weighted policy optimization (AWPO) is an interesting way to balance cloning loss and the reduced tree search (the optimistic value).

**Weaknesses:**

Ultimately, the method gains in learning efficiency for the game studied, which is an important contribution, although it isn’t clear that there is any performance gain compared to other CDTE methods.

The method seems to require global reward information at each agent during execution.

**Questions:**

It seems that the method assumes that the agents have access to the global reward at each step during inference?

In Figure 6, please clarify the difference between communication and sharing.

Section 3.1, the discussion about the degree to which a homogeneous solution is a good one is interesting and certainly depends on the particular scenario, but it isn’t clear how the MAZero fits into this.

Could you say more about the Shared Individual Dynamic Network g ?  Perhaps it is just the terminology but the idea seems confusing.

Section B.1, could you say more about how to “use positional encoding to distinguish agents in homogeneous settings”?

Appendix C, why do you think Adam is more effective than stochastic gradient descent?  Isn't this well known?

Some small items:  Perhaps “ingenious” and “audacious” are terms better left for the reader to decide for themselves?  The paper refers to “real world” cases but ultimately this is about a game environment.

---

> ### Author Response · Authors · 2023-11-22
>
> Thank you very much for your positive recommendation of our work and your insightful comments. Following are our responses to your concerns.
>
> **Q1: It seems that the method assumes that the agents have access to the global reward at each step during inference?**
>
> The MAZero algorithm is designed under the CTDE framework, which means the global reward allows the agents to learn and optimize their policies collectively during centralized training. The predicted global reward is used in MCTS planning (inference) to search for a better policy based on the network prior. Ablation study in Appendix C, Table 8 shows that agents maintain comparable performance using the final model without global reward, communication or planning.
>
> **Q2: In Figure 6, please clarify the difference between communication and sharing.**
>
> We have provided a clearer explanation of the differences between communication and sharing in Figure 6.
>
> - Communication: This refers to the exchange of information between agents to facilitate cooperation. In the MAZero algorithm, communication is achieved through an additional communication block using the attention mechanism. This allows agents to share their individual latent states and promote cooperation during the model unrolling process. “No communication” means the Centralized Dynamic Block in Figure 2 only contains the Shared Individual Dynamic Network g.
> - Sharing: This refers to the parameter sharing among agents in the centralized-value, individual-dynamic model, which is a common technique for multi-agent reinforcement learning. By sharing parameters, the model can capture the homogenous behavior of agents and improve learning efficiency. “No sharing” means there is no parameter sharing in representation function h, dynamic function g and policy prediction function P in Equation (4), i.e., agents learn network individually.
>
> **Q3: Section 3.1, the discussion about the degree to which a homogeneous solution is a good one is interesting and certainly depends on the particular scenario, but it isn’t clear how the MAZero fits into this.**
>
> MAZero fits into the discussion of homogeneous solutions through parameter sharing, which has underscored the huge success in model-free MARL methods. More specifically, agents share representation network, policy prediction network and individual dynamic network, thus exhibit homogeneous behavior when facing the same situation or input similar observations.
>
> **Q4: Could you say more about the Shared Individual Dynamic Network $g_\theta$ ? Perhaps it is just the terminology but the idea seems confusing.**
>
> We have offered additional clarification on the Shared Individual Dynamic Network $g_\theta$ and its role in the model in the revised version. The Shared Individual Dynamic Network $g_\theta$ is responsible for deriving the subsequent local latent state $s_{t,k+1}^i$ of each agent $i$ based on its individual current latent state $s_{t,k}^i$ , individual action $a_{t,k}^i$, and communication feature $e_{t,k}^i$. The term "Shared Individual Dynamic" refers to the combination of shared parameters among agents (to capture homogenous behavior) and individual dynamic modeling for each agent (to capture local information input).
>
> **Q5: Section B.1, could you say more about how to “use positional encoding to distinguish agents in homogeneous settings”?**
>
> Positional encoding is a detailed implementation of the self-attention block in the Communication network $e$. Using positional encoding to distinguish agents in homogeneous settings means that the algorithm incorporates the relative positions of agents in the environment to differentiate between them. This helps the model to learn distinct communication features for each agent, even when their actions or observations are similar.
>
> **Q6: Appendix C, why do you think Adam is more effective than stochastic gradient descent? Isn't this well known?**
>
> While it is well-known that Adam often performs better than stochastic gradient descent (SGD) in many cases, especially for multi-agent reinforcement learning[1], muzero-based algorithms tend to use SGD optimizer[2-4]. So we provide the ablation study to demonstrate the effectiveness of Adam optimizer specifically for our MAZero algorithm. The results show that Adam leads to superior performance and more consistent stability in the experiments.
>
> [1] Yu, Chao, et al. "The surprising effectiveness of ppo in cooperative multi-agent games." *Advances in Neural Information Processing Systems* 35 (2022): 24611-24624.
>
> [2] Schrittwieser, Julian, et al. "Mastering atari, go, chess and shogi by planning with a learned model." *Nature* 588.7839 (2020): 604-609.
>
> [3] Hubert, Thomas, et al. "Learning and planning in complex action spaces." *International Conference on Machine Learning*. PMLR, 2021.
>
> [4] Ye, Weirui, et al. "Mastering atari games with limited data." *Advances in Neural Information Processing Systems* 34 (2021): 25476-25488.

---

### Author Response · Authors · 2023-11-23
**Author Response Summary**

We would like to express our sincere gratitude to all the reviewers for their thorough examination of our paper and their valuable feedback, which we greatly appreciate. In summary we have taken into account the following concerns raised by the reviewers and have provided substantial evidence to address them.

1. **Possible Extensions:**
    - Discussed potential extensions of our proposed algorithm to the single-agent setting, continuous action space setting, and stochastic environments.
    - Emphasized the significance of search-based algorithms and their breakthroughs in various application domains, highlighting our method as a foundation for further exploration in multi-agent settings.
2. **Additional Experiments and Explanations:**
    - Included more baseline algorithms (RODE and CDS) for comprehensive comparisons and added additional random seed data to ensure the robustness of our results.
    - Evaluated our method MAZero on a new multi-agent environment (*academy_3_vs_1_with_keeper*, Google Research Football) to demonstrate its ability to solve stochastic tasks.
    - Conducted an ablation study of OS($\lambda$) and AWPO on single-agent environments (LunarLander and MuJoCo) to showcase the generality of these techniques.
    - Provided an extensive ablation study comparing MAZero and Sampled MuZero on SMAC scenarios to highlight the impact of network design and MCTS improvements.
    - Added an ablation study on decentralized execution in SMAC environments to illustrate comparable performance of individual policy prediction without global reward, communication, or MCTS planning.
3. **Clarification about Details in the Paper:**
    - Enhanced explanations of the network structure, parameter-sharing, communication mechanism, optimistic search, and experimental results to further clarify the extension from MuZero to multi-agent settings.
    - Provided detailed code implementation and hyper-parameters used for each baseline algorithm.

We sincerely appreciate the support from reviewers MHMy and efig after the discussion, and we kindly request the other reviewers to review our responses and consider supporting this work if we have adequately addressed their concerns. We remain open to further discussions if any additional concerns arise. If you have any further questions or need additional assistance, please let us know.

---

### Meta-Review · Area_Chair_svDK · 2023-12-09

**Metareview:**

The paper proposes MAZero, a novel multi-agent reinforcement learning (MARL) algorithm that extends the MuZero framework into multi-agent settings. MAZero integrates Monte Carlo Tree Search (MCTS) with a centralized model, and introduces two new techniques: Optimistic Search Lambda (OS(λ)) and Advantage-Weighted Policy Optimization (AWPO), aiming to improve sample and computational efficiency in large-scale decision-making tasks.

## Strengths:
- Novelty and Technical Contribution: The paper presents a significant advancement in MARL by adapting MuZero, a model-based single-agent RL algorithm, to multi-agent scenarios. The introduction of OS(λ) and AWPO is innovative, addressing the challenges of large action spaces in multi-agent systems.
- Experimental Results: The paper demonstrates MAZero’s effectiveness through extensive experiments on the SMAC benchmark, showing superior sample efficiency compared to model-free approaches and comparable or better performance against existing model-based methods.
- Clarity and Presentation: The paper is well-written with a clear exposition of the algorithm, its theoretical underpinnings, and experimental setup.

## Weaknesses:
- Limited Scope of Environments: The primary experiments are conducted on SMAC environments, which are mostly deterministic. This raises questions about the algorithm’s effectiveness in stochastic environments, a concern partially addressed by supplementary experiments.
- Scalability and Generalizability: There are concerns about the scalability of the proposed techniques to environments with even larger action spaces and how generalizable these techniques are to other types of multi-agent environments.
- Comparative Analysis: The paper could benefit from a broader comparison with a wider range of recent MARL methods. Also, the reliance on certain benchmarks and the lack of utilization of more updated or diverse benchmarks (like SMACv2) could limit the understanding of the algorithm’s full potential.

## Reviewer Comments and Author Responses:
- Reviewers appreciated the technical contribution and the experimental validation of MAZero. However, some pointed out the need for broader testing beyond the deterministic SMAC environments and questioned the algorithm’s applicability to more diverse and stochastic settings.
- The authors responded to these concerns by conducting additional experiments in environments like LunarLander, MuJoCo, and Google Research Football. They also provided more detailed explanations and clarifications on the algorithm’s design and its applicability to different settings.

**Justification For Why Not Higher Score:**

Primarily due to its limited testing in deterministic environments and lack of broader benchmarking and theoretical justification for its novel techniques.

**Justification For Why Not Lower Score:**

Based on the strengths of the paper in advancing MARL and the responses to the concerns raised, the final recommendation is for acceptance as a poster presentation. The paper contributes significantly to the field of MARL with its novel approach and thorough experimentation. Future work should focus on addressing the scalability in more diverse and stochastic environments to enhance the generalizability of the proposed techniques.

---

### Decision · Program_Chairs · 2024-01-16

Accept (poster)